# Entire expressed peripheral blood transcriptome in pediatric severe malarial anemia

Samuel B. Anyona [1,2] ✉, Qiuying Cheng [3], Sharley A. Wasena[2,4], Shamim W. Osata [2], Yan Guo[5], Evans Raballah [2,6], Ivy Hurwitz [3], Clinton O. Onyango [2,4], Collins Ouma [2,4], Philip D. Seidenberg[7], Benjamin H. McMahon[8], Christophe G. Lambert [9], Kristan A. Schneider [9,10] & Douglas J. Perkins [2,3] ✉

This study on severe malarial anemia (SMA: Hb < 6.0 g/dL), a leading global cause of childhood morbidity and mortality, compares the entire expressed whole blood host transcriptome between Kenyan children (3-48 mos.) with non-SMA (Hb ≥ 6.0 g/dL, $n = 39$) and SMA ($n = 18$). Differential expression analyses reveal 1403 up-regulated and 279 down-regulated transcripts in SMA, signifying impairments in host inflammasome activation, cell death, and innate immune and cellular stress responses. Immune cell profiling shows decreased memory responses, antigen presentation, and immediate pathogen clearance, suggesting an immature/improperly regulated immune response in SMA. Module repertoire analysis of blood-specific gene signatures identifies up-regulation of erythroid genes, enhanced neutrophil activation, and impaired inflammatory responses in SMA. Enrichment analyses converge on disruptions in cellular homeostasis and regulatory pathways for the ubiquitin-proteasome system, autophagy, and heme metabolism. Pathway analyses highlight activation in response to hypoxic conditions [Hypoxia Inducible Factor (HIF)−1 target and Reactive Oxygen Species (ROS) signaling] as a central theme in SMA. These signaling pathways are also top-ranking in protein abundance measures and a Ugandan SMA cohort with available transcriptomic data. Targeted RNA-Seq validation shows strong concordance with our entire expressed transcriptome data. These findings identify key molecular themes in SMA pathogenesis, offering potential targets for new malaria therapies.

Malaria remains a significant global public health challenge with 249 million annual cases and 608,000 deaths[1]. The majority of the cases (233 million, 93.6%) and mortality (580,000, 95.4%) occurred in the WHO African region and are due to infections with *Plasmodium falciparum*[1], mainly in children under five years of age. Kenya faces a substantial challenge with *P. falciparum* malaria, reporting ~3.42 (2.23–5.02) million annual cases and ~11,788 (11,100–12,700) deaths,

primarily in the under-five population[1]. Since the disease burden increases with transmission intensity, severe malaria remains among the leading causes of morbidity and mortality in children residing in holoendemic *P. falciparum* regions of Kenya, and other such regions of sub-Saharan Africa[2–4]. In high transmission regions, the primary manifestation of severe malaria is severe malarial anemia [SMA, hemoglobin (Hb)<6.0 g/dL] in the presence and absence of

respiratory distress, with cerebral malaria occurring only in rare (atypical) cases[2,5,6].

The etiology of SMA is multifaceted and includes overlapping characteristics such as the lysis of infected and uninfected erythrocytes, sequestration of erythrocytes in the spleen, and suppression of bone marrow functions[7,8]. Although natural immunity is acquired following repeated infections with *P. falciparum*[9–11], innate immunity is particularly important for influencing disease severity in young, malaria-naïve children. Our previous longitudinal studies in Kenyan children using a combination of candidate-gene approaches, genome-wide association studies, high-throughput genotyping, and array-based whole transcriptional profiling revealed that the development of SMA is mediated, partially by innate immune response genes[7,12–15]. Our targeted transcriptome analyses also revealed that differentially expressed genes (DEGs) in host ubiquitination processes are a central feature of SMA pathogenesis[16]. Microarray analysis of candidate genes in whole blood identified DEGs that encode amino acid transport, phospholipid metabolic processes, and positive regulation of nitrogen metabolism in Gabonese children with SMA (<6 years old)[17].

Advances in high-throughput sequencing technologies and bioinformatics analyses have provided important insights into the human immune response to *P. falciparum* and identified potential vaccine candidates[17–20]. For example, studies investigating gene expression in adults during controlled human malaria infection experiments identified >2,700 DEGs in the whole blood transcriptome, for which a subset of 265 genes was associated with transcription and cell-cycle regulation, phosphatidylinositol signaling, and erythrocytic development[21]. In addition, next-generation RNA sequencing in whole blood, which has the potential to concomitantly capture host and parasite gene expression, showed that severe malaria (i.e., cerebral malaria, hyperlactatemia, or their combination) in Gambian children (<16 years old) was associated with increased expression of granulopoiesis and interferon-γ–related genes, and inadequate suppression of type 1 interferon signaling[18]. Additional analysis of whole blood transcriptomes in Ugandan children (age; 18 mos. to 12 yrs.) revealed that erythropoietic and nuclear factor erythroid 2 like 2 (NRF2)-regulated genes were differentially expressed between cerebral malaria and SMA cases[22]. Despite progress in defining the human immune response to *P. falciparum*, a comprehensive investigation of the entire expressed transcriptome has not been reported in children who develop SMA, the group that suffers the highest global morbidity and mortality[2,5,8,23–25]. Here, we present the top emergent biological processes, networks, and pathways for the first entire expressed whole blood transcriptome in Kenya children (<5 years old) from a holoendemic region of western Kenya who develop SMA as the exclusive phenotype of severe malaria.

## Results

### Demographic and clinical characteristics of the study participants

Since anemia definitions, particularly in pediatrics, require georeferencing[26], we utilized data from our cohort of 1644 children (3-48 mos) followed over a 36-month period for which we recorded over 19,000 hemoglobin measurements. Because childhood SMA is among the leading causes of anemia-related death in western Kenya, we utilized a dynamic programming algorithm to determine average Hb concentrations across 36 months of follow-up to define categories that best-captured childhood mortality[27]. This method revealed three distinct Hb categories at $p < 0.001$: ≤5.90 g/dL ($n = 62$, mortality fraction = 0.53), 5.91–8.09 g/dL ($n = 209$, mortality fraction = 0.15), and >8.09 g/dL ($n = 1373$, mortality fraction = 0.03, Supplementary Fig. S1). These results parallel those in a previous longitudinal birth cohort (0-48 mos.) with 14,317 repeated Hb measurements from the same geographical region[28]. Based on this data-driven approach, two groups were investigated: SMA defined as Hb<6.0 g/dL and non-SMA as Hb ≥

6.0 g/dL. For the RNA-Seq analysis, 39 children (3–48 mos.) with non-SMA and 18 with SMA were selected from the overall cohort ($n = 577$, 1–59 mos.), excluding those with HbSS (Supplementary Fig. S2). Admission demographic and clinical characteristics of the children selected for RNA-Seq in whole blood are shown in Table 1. Based on the selection criteria, sex ($p = 0.777$), overall age ($p = 0.283$), and distribution within age categories ($p = 0.781$) were comparable. Glucose levels ($p = 0.377$) and axillary temperature ($p = 0.121$) were also comparable between the groups. Consistent with more profound anemia in children with SMA, hematocrit ($p = 1.678E−09$) and red blood cell counts ($p = 2.072E−09$) were lower. White blood cell ($p = 0.007$) and lymphocyte ($p = 0.001$) counts were also elevated in children with SMA. Other hematological measures were comparable between the groups, as were parasitological indices, clinical features, and genetic variants.

### Comparative cellular pathways in non-SMA and SMA reveals divergent metabolic and immune responses

To identify unique and shared genes, a Venn diagram analysis was performed on the 53,286 transcripts, revealing 602 genes that were uniquely expressed in non-SMA, 493 in SMA, and 16,036 co-expressed genes (Fig. 1a). Uniquely expressed genes in non-SMA and SMA were then explored by canonical pathway analysis (MetaCore™), revealing four significant sub-networks (Supplementary Table S1). The top-ranked sub-network in children with non-SMA was [TFF3↔IL-6↔IL6RA↔ADAM17↔gp130, ($p = 3.180E−12$)], highlighting gene ontology (GO) processes for the crucial role of T-helper 17 cell lineage commitment and differentiation, interleukin-6-mediated signaling, and T-helper cell lineage commitment in driving T-helper 17 type immune responses. The second sub-network [SHOX2↔Neuregulin1↔FGFR2↔Endothelin-1↔ECE2, ($p = 1.930E−07$)] indicates processes that regulate cellular signaling and proliferation, particularly through receptor tyrosine kinase and enzyme-linked pathways. The top-ranked sub-network in children with SMA [c-Myc↔C/EBPbeta↔STAT1↔STAT5 ↔ERK1/2, ($p = 1.240E−200$)] was associated with signaling pathways linked to growth factors, peptide hormones, stimuli, and transmembrane receptor protein tyrosine kinases. The second sub-network [ACE1↔des-Arg9-bradykinin↔BDKRB1↔des-Arg10-kallidin↔BDKRB2, ($p = 5.310E−103$)] underscores positive regulation of cellular responses associated with hormones, chemicals, and metabolic processes. Collectively, these data indicate that non-SMA was characterized by changes in immune response and cell signaling, while SMA involved a broader spectrum of cellular activities, including metabolic regulation and hormone responses. Differential expression analysis of the 53,286 transcripts was then performed, identifying 1682 DEGs [false discovery rate (FDR)-adjusted, $p$adj < 0.050]: 1403 up- and 279 down-regulated genes in children with SMA (Fig. 1b).

### Insights on gene expression variance and group heterogeneity

A PCA analysis revealed that the first principal component (PCA1) accounted for 36.1% of total variance and differentiated a portion of the non-SMA and SMA groups, while PCA2 (13.8% total variance) had tighter clustering for the non-SMA group (Supplementary Fig. S3). The non-SMA group displayed tighter clustering (homogeneity) across age, sex, and sickle cell traits. In contrast, the SMA group exhibited more dispersion, notably among males and those with the HbAA trait. Thus, while there were distinct gene expression features that distinguished between the two groups, children with SMA were more heterogeneous.

### Functional interactions and pathway networks indicate altered immune responses, metabolic processes, and erythrocyte differentiation

To further explore the pathogenesis of SMA, we performed non-supervised hierarchical cluster analysis of the significant DEGs ($n = 1682$). These analyses identified three unique co-regulated gene

**Table 1 | Demographic and clinical characteristics of the study participants**

| Characteristics | Total | non-SMA (Hb≥6.0 g/dL) | SMA (Hb<6.0 g/dL) | p-value |
|---|---|---|---|---|
| No. of participants, n | 57 | 39 | 18 | |
| **Sex, n (%)** | | | | |
| Male | 29 (50.9) | 19 (48.7) | 10 (55.6) | 0.777[a] |
| Female | 28 (49.1) | 20 (51.3) | 8 (44.4) | |
| **Age, months,** | 21.0 (21.5) | 23.0 (21.0) | 16.0 (22.8) | 0.283[b] |
| 0–12.9 | 12 (21.1) | 7 (17.9) | 5 (27.8) | 0.781[a] |
| 13–24.9 | 21 (36.8) | 14 (35.9) | 7 (38.9) | |
| 25–35.9 | 12 (21.1) | 9 (23.1) | 3 (16.7) | |
| 36–48.9 | 12 (21.1) | 9 (23.1) | 3 (16.7) | |
| Glucose, mmol/L | 5.1 (2.1) | 5.0 (2.3) | 5.3 (1.8) | 0.377[b] |
| Admission temperature, °C | 38.0 (1.2) | 38.0 (1.0) | 37.8 (0.7) | 0.121[b] |
| **Hematological Parameters** | | | | |
| Hemoglobin, g/dL | 9.3 (5.2) | 9.9 (1.3) | 4.7 (0.9) | NA |
| Hematocrit, % | 28.0 (16.4) | 30.2 (5.7) | 14.6 (2.6) | 1.678E-09[b] |
| Red blood cells, × $10^6$/µL | 3.9 (2.4) | 4.3 (1.0) | 2.0 (0.5) | 2.072E-09[b] |
| Red cell distribution width, % | 19.4 (4.1) | 18.7 (2.7) | 20.6 (5.5) | 0.032[b] |
| Mean corpuscular volume, fL | 69.7 (11.8) | 69.5 (8.6) | 73.5 (15.4) | 0.105[b] |
| Mean corpuscular hemoglobin, pg | 23.3 (5.4) | 22.9 (4.0) | 23.8 (7.9) | 0.381[b] |
| Mean corpuscular hemoglobin concentration, g/dL | 32.3 (6.4) | 32.5 (6.8) | 31.4 (5.5) | 0.744[b] |
| Platelets, ×$10^3$/µL | 120.0 (93.4) | 120.0 (82.4) | 121.5 (110.0) | 0.687[b] |
| Platelet distribution width, % | 16.5 (1.2) | 16.5 (1.3) | 17.2 (1.2) | 0.560[b] |
| Mean platelet volume, fL | 8.6 (1.7) | 8.5 (1.5) | 9.0 (1.6) | 0.112[b] |
| White blood cells, ×$10^3$/µL | 12.3 (7.6) | 11.0 (6.8) | 16.7 (11.2) | 0.007[b] |
| Lymphocytes, ×$10^3$/µL | 3.9 (3.4) | 3.6 (1.9) | 6.7 (6.9) | 0.001[b] |
| Monocytes, ×$10^3$/µL | 1.4 (1.3) | 1.2 (1.2) | 1.6 (1.5) | 0.163[b] |
| Neutrophils, ×$10^3$/µL | 5.3 (4.9) | 5.3 (3.4) | 5.2 (8.1) | 0.813[b] |
| Granulocytes, ×$10^3$/µL | 7.3 (4.8) | 6.7 (3.5) | 9.9 (8.6) | 0.259[b] |
| **Parasitological Indices** | | | | |
| Parasite density, MPS/µL | 46,398 (82,297) | 60,917 (81,169) | 24,846 (92,647) | 0.471[b] |
| Low (1–5000) | 9 (15.8) | 5 (12.8) | 4 (22.2) | 0.281[a] |
| Moderate (5001–50,000) | 20 (35.1) | 12 (30.8) | 8 (44.4) | |
| High (50,001–100,000) | 16 (28.1) | 14 (35.9) | 2 (11.1) | |
| Hyper (>100,001) | 12 (21.1) | 8 (20.5) | 4 (22.2) | |
| Geomean parasitemia, /µL | 29,974 | 31,909 | 26,174 | 0.201[c] |
| **Clinical Features** | | | | |
| Respiratory distress | 5 (8.8) | 1 (2.6) | 4 (22.2) | 0.031[a] |
| Hypoxia, SpO₂ < 90% | 2 (3.5) | 1 (2.6) | 1 (5.6) | 0.519[a] |
| Convulsions | 22 (39.3) | 16 (41.0) | 6 (39.3) | 0.461[a] |
| Hypoglycemia (blood glucose levels <2.2 mM) | 3 (5.4) | 2 (5.1) | 1 (5.9) | 0.670[a] |
| Jaundice | 1 (1.8) | 1 (2.6) | 0 (0.0) | - |

**Table 1 (continued) | Demographic and clinical characteristics of the study participants**

| Characteristics | Total | non-SMA (Hb≥6.0 g/dL) | SMA (Hb<6.0 g/dL) | p-value |
|---|---|---|---|---|
| Thrombocytopenia (platelet count <150×$10^3$/mm³) | 37 (64.9) | 24 (61.5) | 13 (72.2) | 0.317[a] |
| **Genetic Variant** | | | | |
| **Sickle cell trait, n (%)** | | | | |
| Hb AA | 51 (89.5) | 35 (89.7) | 16 (88.9) | 0.923[a] |
| Hb AS | 6 (10.5) | 4 (10.3) | 2 (11.1) | |

[a]Fisher's exact test [presented as number (%)] with exact p-values for homogeneity was performed.
[b]Two-sided Mann-Whitney-U tests [presented as median (IQR)] were used to compare the non-SMA and SMA groups.
[c]Group means were compared by two-sample t-test [presented as mean (SEM)], with equal variance.
Data are presented as number (percentages; %), median (interquartile range; IQR) or mean (standard error of mean; SEM) unless otherwise noted. Children (n = 57) presenting with malaria at SCRH were recruited. Based on hemoglobin (Hb) levels, children were categorized into either non-severe malaria anemia (non-SMA; Hb≥6.0 g/dL, n = 39) or severe malarial anemia (SMA; Hb < 6.0 g/dL, n = 18).
All p-values shown in bold remained below the significance level after multiple test corrections using the Bonferroni-Holm method (familywise error rate, significance level at 0.050). For the clinical features, none of the children had spontaneous bleeding. MPS—malaria parasites presented as mean (standard deviation).

clusters that differentiated children with non-SMA and SMA with distinct patterns for WBCs and lymphocytes but not for parasitemia levels, sickle cell status (HbAA vs. HbAS), or age distribution (Fig. 1c). To gain further insight into the networks for the major co-regulated gene clusters, canonical pathway maps for direct functional interactions were generated. The network for down-regulated genes (n = 279, cluster 1) in SMA was IRF1↔SUZ12↔IL-1β↔NRF2↔LHX2 with the transcription factor, IRF1, as the central divergence hub (green-square box, 56 direct interactions) and IL-1β as the central convergence hub (red-square box, 44 direct interactions, Fig. 1d). The top processes associated with the functional interactions for cluster 1 were regulation of innate immune response (p = 3.272E−17), and defense response (p = 6.348E−17). Cluster 2 (n = 489, up-regulated) generated a TAL1↔LYL1↔BRD4↔FOXO3A↔EKLF1 network with the transcription factor, TAL1, as the central divergence hub (green-square box, 343 direct interactions). The central convergence hub in cluster 2 (red-square boxes) was the transcription factor E2F2 (7 functional interactions), while the secondary divergence hubs were GLUT1 and HMBS (both with 7 direct interactions, Fig. 1e). Top interactions for cluster 2 were metabolic process (p = 1.804E−25) and erythrocyte differentiation (p = 1.425E−13). Collectively, SMA was associated with suppressed innate immune stress responses, elevated metabolic processes, and promotion of erythrocyte differentiation.

**Altered leukocytic immune cell profiles in SMA**
To determine if leukocytic immune profiles differed in children who developed SMA, a bioinformatics approach was implemented using CIBERSORTx[29,30]. Despite interindividual variability, transcriptional profiling identified five immune cell types that were differentially expressed at p < 0.050 (Fig. 2a). Children with SMA had increased expression of naïve B cells (p = 9.741E-05) and CD8 T cells (p = 0.026) (Fig. 2b). In contrast, the SMA group had a lower proportion of expression for memory B cells (p = 0.035), activated dendritic cells (p = 0.007), and neutrophils (p = 0.026) (Fig. 2b). Based on the immune cell type profiles in the expression data, children with SMA appeared to have a decreased antigenic response and reduced immune priming.

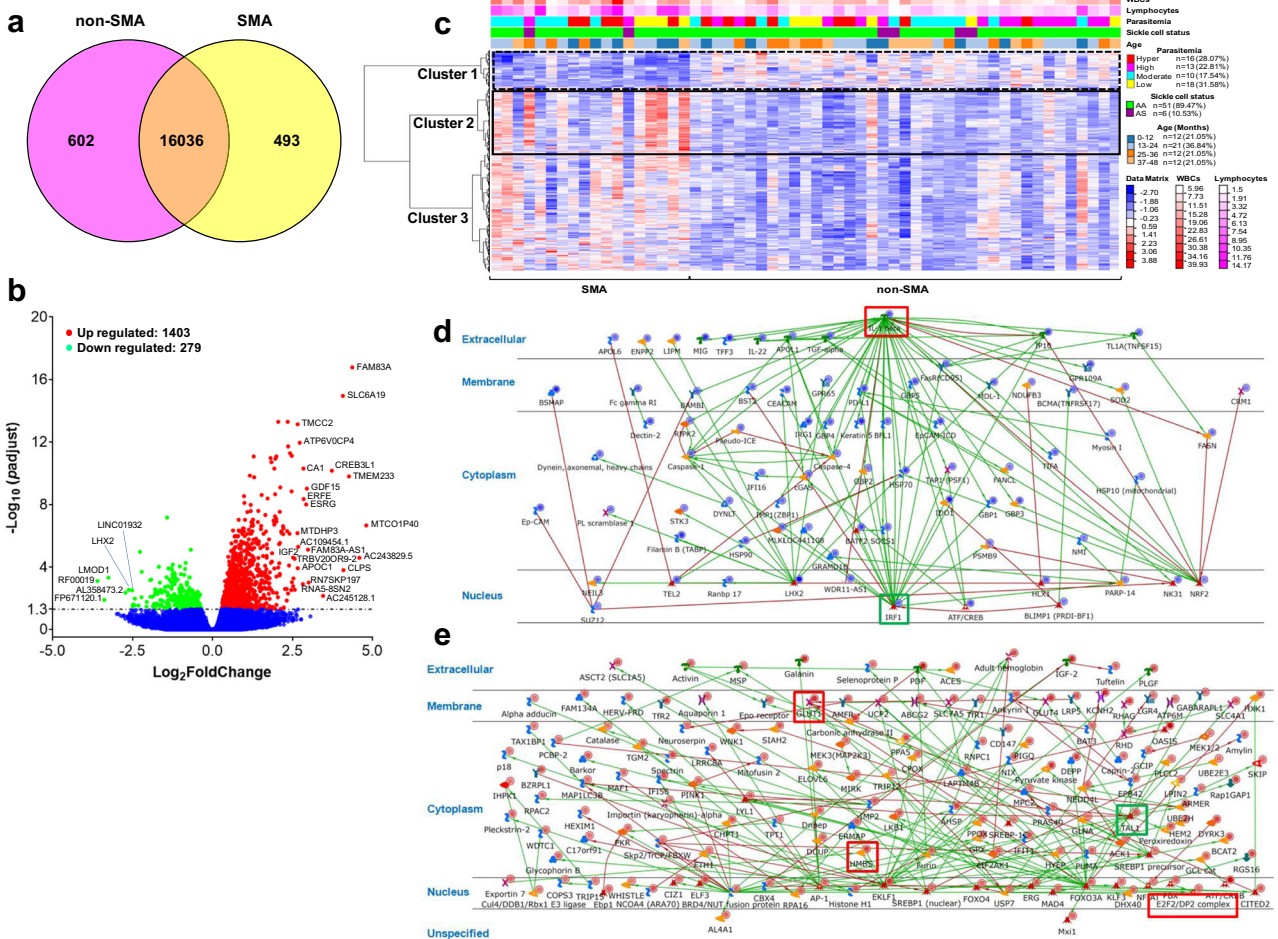

**Fig. 1 | Differential gene expression analysis in children with severe malaria anemia.** EdgeR (3.16.5) was used to infer the overall distribution of differentially expressed genes. **a** Venn Diagram depicting relationship (similarities and differences) of 53,286 transcripts between the clinical groups, with unique and co-expressed genes within each group shown, and overlapping regions indicating shared gene expression patterns. **b** Volcano plot showing 1,403 up-regulated and 279 down-regulated protein-coding genes in Kenyan children presenting with non-SMA (Hb≥6.0 g/dL, $n$ = 39) and SMA (Hb<6.0 g/dL, $n$ = 18). The significance measure (exact-test based on the negative binomial distribution) is shown on the Y-axis as negative logarithm of $p$-adjusted (-$\log_{10}(p$adj-value), and the effect size is depicted on the X-axis [$\log_2$(FoldChange)]. Significance set at $p$adj < 0.050 and $\log_2$(FoldChange)>1.3. Blue dots represent genes with no significant difference, red dots denote up-regulated genes, and green dots represent down-regulated genes. **c**. Hierarchical Clustering Heatmap of 1,682 differentially expressed genes, illustrating clustering analysis based on $\log_2$(FPKM + 1) values. WBCs counts, lymphocytes, parasitemia, sickle cell trait status, and age distributions are shown. Clusters 1 (down-regulated in SMA) and 2 (up-regulated in SMA) are delineated with hatched and solid black outlines, respectively. **d, e** DEGs enrichment analysis of process networks based on clusters from the heatmap, highlighting significant gene networks and central divergence/convergence hubs. Blue circles represent down-regulated genes, while red circles denote up-regulated genes. The IRF1 ↔ SUZ12 ↔ IL-1β ↔ NRF2 ↔ LHX2 network contained 279 down-regulated genes (Cluster 1) with the transcription factor, IRF1, as the central divergence hub (green box) and IL-1β as the central convergence (red box, Fig. 1d). The TAL1 ↔ LYL1 ↔ BRD4 ↔ FOXO3A ↔ EKLF1 network shows 489 up-regulated genes (Cluster 2) with TAL1 as the central divergence hub (green box) and the transcription factor, E2F2, as the central convergence hub containing secondary convergence hubs, GLUT1 and HMBS (red boxes, Fig. 1e). Details of symbols used in these figures are available at: https://portal.genego.com/legends/MetaCoreQuickReferenceGuide.pdf.

## Blood transcriptome module repertoire analysis highlights erythroid upregulation and neutrophil activation amidst impaired inflammation in SMA

To further identify host immunological profiles associated with SMA, blood-specific gene expression signatures were examined using the BloodGen3Module in R[31,32]. To focus on the most enriched signals, the top 10% of the up- and down-regulated gene clusters are presented. As shown in Fig. 2c, d, and Supplementary Table S2, up-regulated gene patterns were largely encompassed by erythroid cells (e.g., *HBBP1*, *RHD*, *ST3GAL1*, *SLC2A1*, and *GCLC*), protein modification (e.g., autophagy and ubiquitin-proteome system: *FURIN*, *GABARAPL2*, *TM7SF2*, *RNF14*, *UBE2M*, *TAX1BP1*), and neutrophil activation (e.g., *CEACAM6*, *CEACAM8*, *CTSG*, *DEFA3*, DEFA4). Conversely, the top down-regulated gene signatures were for inflammation (e.g., *FCGR3B*, *TLR1*, *TLR5*, *FPR2*, *HIF1A*, *SOD2*, *HSD17B11*, *MAPK14*, *SOCS3*, *LIMK2*, *ACSL1*, *NDUFB3*),

interferon and cytokines/chemokines (*CSF2RB*, *FPR2*, *IFIT2*, *IFIT5*, *MX1*, *NOD2*, *GBP1*, *GBP4*, *GBP5*, *CASP1*, *CASP4*, *CASP5*, *MAPK14*, *TAP1*, *PSMB9*, *SP100*), and neutrophils (e.g., *HSPA1A*, *ADAM8*, *FCGR2A*, *FCGR3B*, *FPR1*, *CYB5R4*, *NCF4*, *HLA-E*, *LCP1*, *BCL3*, *IRF1*, *NFKBIA*, *BCL3*, and *MCL1*, Fig. 2c, d, and Supplementary Table S2). Individual-level data for the top aggregates and accompanying gene expression modules are shown in Fig. 2e. In summary, BloodGen3Module analysis showed that SMA was characterized by the upregulation of erythroid cells, enhanced neutrophil activation, and impaired inflammatory response.

## Functional Enrichment Analysis Reveals Disruptions in Protein Degradation, Heme metabolism, Cellular Clearance Mechanisms, and Efferocytosis in SMA

To identify characteristics associated with developing SMA, Gene Ontology (GO) enrichment analysis was performed for three domains

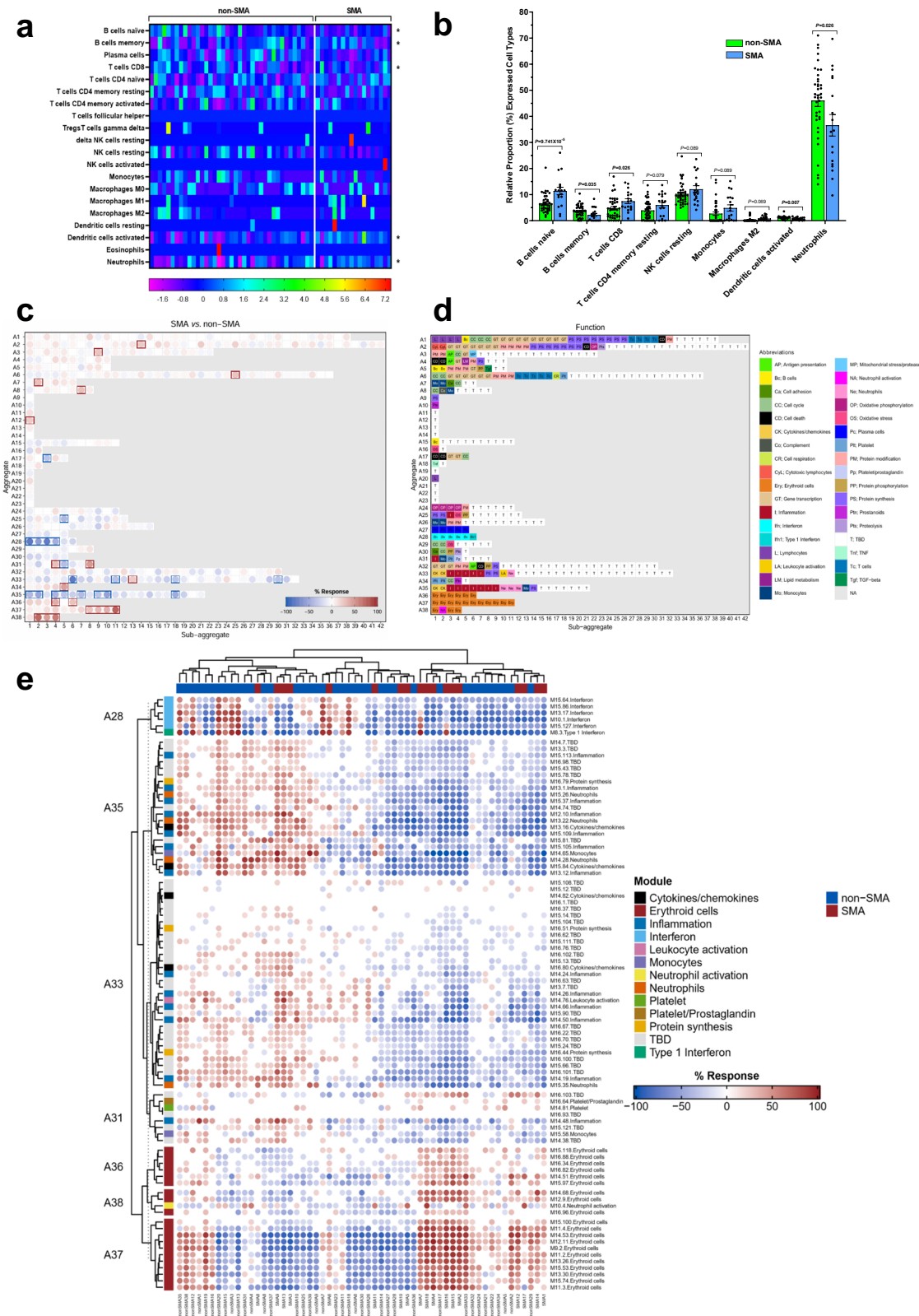

(i.e., biological process, cellular components, and molecular functions) with the top 20 in each presented in Fig. 3a. Of the three domains, biological processes exhibited the most significant enrichment in children with SMA. For example, proteasome-mediated ubiquitin-dependent protein catabolic process ($p$adj = 8.392E−05), proto-porphyrinogen IX metabolic process ($p$adj = 1.266E−04), and regulation of autophagy ($p$adj = 1.275E−04) were the top-ranked biological processes. The ubiquitin ligase complex ($p$adj = 2.659E−03), phago-phore assembly site ($p$adj = 6.669E−03), and hemoglobin complex ($p$adj = 6.804E−03) were the top cellular components enriched in SMA. The chromatin DNA binding ($p$adj = 5.046E−04), protein-macromolecule adaptor activity ($p$adj = 6.103E−03), and transcription corepressor activity ($p$adj = 6.103E−03) showed the greatest enrich-ment for molecular functions. Additional analysis using the Reactome

**Fig. 2 | Cellular composition and module repertoire analysis of whole blood.**
Deconvolution analysis of the different cell types in blood was determined using CIBERSORTx. Cellular frequencies were imputed using LM22 as the signature matrix file. **a** Heatmap representing the cell type expression for 22 types/subtypes of leukocyte cell populations presented at the individual patient level in the non-SMA (Hb≥6.0 g/dL, $n = 39$) and SMA (Hb<6.0 g/dL, $n = 18$) groups. An asterisk (*) indicates significant differences in immune cell proportions between the two groups determined using two-sided, two-sample $t$-tests with Welch correction, and $p < 0.050$. **b** Relative proportion (%) of expression for immune cell types differing between non-SMA (n = 39) and SMA ($n = 18$) groups, presented as mean (SEM) after bivariate analysis using two-sided, two-sample $t$-tests with Welch correction. Module repertoire analysis of DEGs was performed using the BloodGen3Module R package. **c** Module fingerprint grid plot analysis results following Welch's-corrected

$t$-test. Each module is positioned on a grid, with rows corresponding to 'module aggregates' reflecting similar abundance patterns across reference datasets of distinct immune states. The number of constitutive modules for each aggregate varies from 1 (A9-A14, A19-A23) to 42 (A2). Red spots indicate "up-regulated modules", while blue spots represent "down-regulated modules" in children with SMA relative to non-SMA. **d** Biological/immunological function associated with each of the modules within the grid. **e** Fingerprint heatmap represents patterns of annotated modules across individual study participants. The heatmap displays the abundance patterns of 85 annotated modules using an FDR correction with a 20% differentially expressed significance level. Hierarchical clustering arranges samples (columns) and modules (rows), with color gradients indicating proportions of differentially expressed transcripts ranging from blue (decreased) to red (increased).

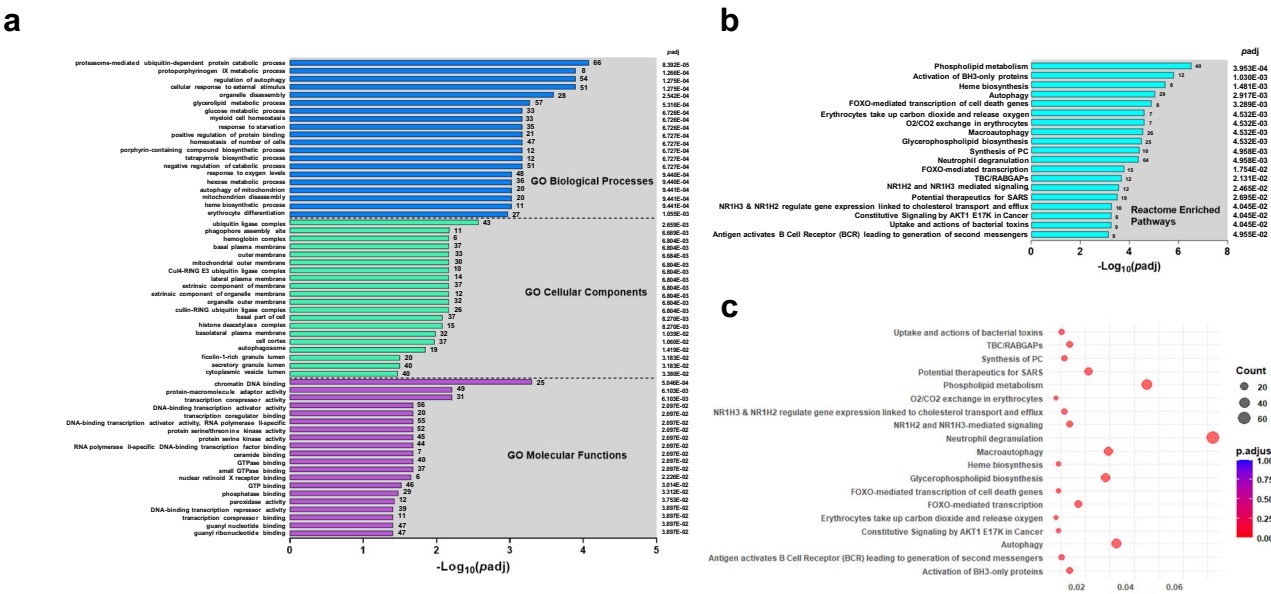

**Fig. 3 | Functional enrichment analysis of differentially expressed genes in severe malarial anemia. a** Gene Ontology (GO) Enrichment Analysis presenting the top 20 enriched terms in biological process, cellular component, and molecular function categories of DEGs in children with SMA (Hb<6.0 g/dL, $n = 18$) compared to non-SMA (Hb≥6.0 g/dL, $n = 39$) group. Enrichment analysis was conducted using the clusterProfiler R package, correcting for gene length bias. Enriched GO terms were determined by hypergeometric test, with $p$-adjusted values < 0.050 considered significant. The X-axis represents the negative logarithm of $p$-adjusted (-Log$_{10}$[$p$ adjusted]) values. Significance determined by one-sided over-representation analysis with multiple testing corrections using the Benjamini-

Hochberg procedure. **b** Reactome enrichment analysis of 19 enriched terms that were significantly different in children with SMA. Pathway names are represented on the Y-axis, while the X-axis shows the -Log$_{10}$($p$ adj) values at <0.050. Statistical significance was computed using one-sided overrepresentation analysis (correction by Benjamini-Hochberg procedure). **c** Reactome enrichment histogram displaying the emerging 19 terms. Pathway names are represented on the Y-axis, while the X-axis indicates the gene ratio of up-and-down-regulated genes. The size of the black dots corresponds to the number of annotated genes, and the depth of red color indicates the magnitude of enrichment ($p$adj < 0.050).

enrichment identified the phospholipid metabolism ($p$adj = 3.953E-04), activation of BH3-only proteins ($p$adj = 1.030E-03), and heme biosynthesis ($p$adj = 1.481E-03) as the top-ranked pathways in children with SMA (Fig. 3b and c). To complement this analysis, functional classification of the DEGs for the KEGG pathways revealed three significant pathways: efferocytosis ($p$adj = 1.031E-04), mitophagy-animal ($p$adj = 1.839E-03), and autophagy-animal ($p$adj = 1.940E-02, Supplementary Fig. S4). Together, the enrichment analyses indicated that SMA was characterized by a multifaceted disturbance involving protein degradation (ubiquitination), heme metabolism, cellular clearance mechanisms (autophagy), and efferocytosis, consistent with disruptions in cellular homeostasis and regulatory pathways.

**Canonical Pathways Converge on Alterations in Cellular Stress Response, Immune Modulation, and Metabolic Changes in SMA**
Additional characterization of SMA pathogenesis was carried out by exploring the top 10 significant canonical pathway maps with a

Log$_2$FoldChange = 0.585 (1.5 linear FoldChange) and false discovery rate (FDR) < 0.050 (MetaCore™, Fig. 4a). The top-ranked pathways in the functional categories were: (i) apoptosis and survival—Regulation of apoptosis by mitochondrial proteins ($p$adj = 4.150E-02); (ii) immune response—IFN-alpha/beta signaling via MAPKs ($p$adj = 5.665E-03); (iii) oxidative stress—ROS signaling ($p$adj = 2.089E-02); (iv) regulation of metabolism—Glucocorticoid receptor signaling in glucose and lipid metabolism ($p$adj = 2.089E-02); (v) signal transduction—mTORC2 downstream signaling ($p$adj = 2.105E-02); and (vi) transcription—HIF-1 targets ($p$adj = 1.038E-02). These pathways collectively revealed a complex interplay of cellular stress response, immune modulation, metabolic changes, and transcriptional regulation in SMA.

**Validation of whole blood transcriptome data using an external dataset and targeted RNA-Seq**
To validate the RNA-Seq results, we utilized a transcriptomic dataset from a Ugandan cohort of children with SMA ($n = 17$) and community

**a**

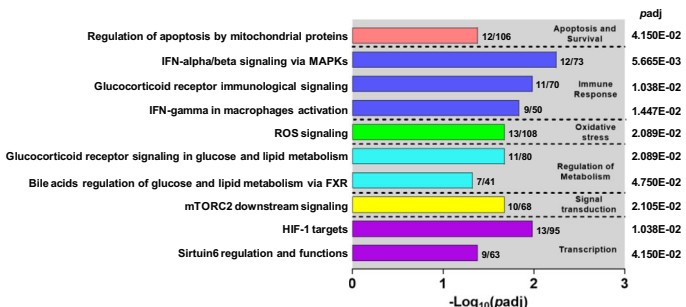

**b**

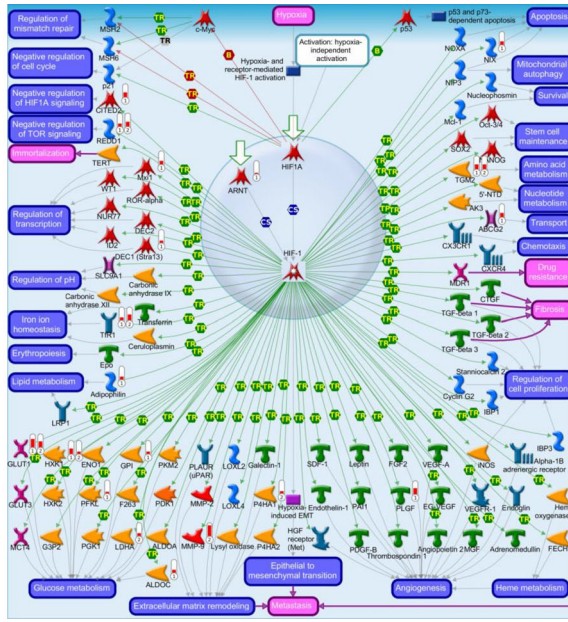

**Fig. 4 | Top-ranked Metacore™ canonical pathway maps in severe malarial anemia. a** The top 10-ranked canonical pathway maps that emerged from the RNA-Seq analysis in SMA (Hb<6.0 g/dL, $n = 18$) compared to non-SMA (Hb≥6.0 g/dL, $n = 39$) according to $p$-adjusted values. The top 10 maps that emerged represent six functional categories: (i) Apoptosis and Survival, (ii) Immune Response, (iii) Oxidative Stress, (iv) Regulation of Metabolism, (v) Signal Transduction, and (vi) Transcription. The left Y-axis indicates the specific biological pathways that were established by non-contradictory state-of-the-art knowledge of the major categories for human metabolism and cell signaling. The right Y-axis shows $p$-adjusted values for each pathway map. The X-axis represents the -Log₁₀($p$adj) value.

Statistical test computed using a hypergeometric probability formula, and $p$adj < 0.050. **b** The second-top-ranked canonical pathway map for Kenyan children [SMA ($n = 18$) versus non-SMA ($n = 39$)] and top-ranked canonical pathway map for Ugandan children [SMA ($n = 17$) versus community children (household controls, $n = 12$)] was Hypoxia-Inducible Factor (HIF)-targets in transcription. The red thermometers indicate annotated genes that were up-regulated in children with SMA (1 = Kenyan children and 2 = Ugandan children), while the blue thermometers indicate down-regulated genes (1 = Kenyan children and 2=Ugandan children). The details of symbols used in these figures are available at: https://portal.genego.com/legends/MetaCoreQuickReferenceGuide.pdf.

children (household controls, $n = 12$)[22]. Enrichment analysis with 323 genes [Log₂FoldChange = 0.585 and false discovery rate (FDR) < 0.050] for pathway maps identified HIF-1 targets involved in transcription as the top emergent pathway ($p$adj = 1.405E−03, Supplementary Fig. S5), the second top-ranked pathway map in our dataset. As such, both datasets were mapped on the HIF-1 targets pathway for a direct comparison of DEGs (Fig. 4b). Of the top 10 pathways in the Ugandan dataset (Supplementary Fig. S5), the only other common pathway in the Kenyan dataset was ROS signaling involved in oxidative stress ($p$adj = 3.994E−02, Supplementary Fig. S6). To further validate DEGs identified in the whole blood transcriptomic analysis, we utilized a Qiagen targeted RNA sequencing panel of 491 immune response genes in a separate cohort of Kenyan children [non-SMA ($n = 23$) and SMA ($n = 20$), Supplementary Table S3]. To enhance biological relevance and reduce noise by excluding non-relevant variations that might be present in either or both datasets, cluster analysis was performed for the genes that were significant ($p < 0.050$) in both datasets ($n = 147$). A heatmap cluster analysis showed concordance in the fold-change and directionality in the two platforms (Fig. 5a) with the two datasets showing a robust correlation ($r = 0.612$; 95% confidence interval (CI), 0.496-0.706; $p = 1.842E−16$, Fig. 5b). The external dataset and targeted RNA sequencing panel validation have high concordance with the whole transcriptomics in Kenyan children, indicating consistent DEGs across different SMA cohorts.

**Comparative analysis of transcriptomics and protein abundance**
To determine if the DEGs from the RNA-Seq align with changes in protein levels, we compared the whole blood transcriptome data with protein abundance (measured in plasma with a 7k SomaScan

platform). MetaCore™ was used to map genes to their respective protein products for 35 children ($n = 19$ non-SMA and $n = 16$ SMA) who had both measurements available. Analysis of the 405 gene/protein pairs ($p < 0.050$) showed concordance in the heatmap (Fig. 6a) and a scatterplot with a modest positive correlation ($r = 0.205$; 95% CI, 0.107-0.299; $p = 3.200E−05$, Fig. 6b). Pathway maps were then generated in MetaCore™, without thresholds, allowing for a comprehensive view of all transcript/protein relationships. The most aligned pathway was HIF-1 targets, showing high significance in both transcripts ($p$adj = 2.391E−21; 95/95 nodes) and proteins ($p$adj = 4.344E−18; 70/95 nodes, Supplementary Fig. S7), indicating a concordant biological overlap between transcriptomic and protein abundance changes, especially for HIF-1-related processes. Collectively, these data demonstrated a modest relationship between the magnitude and directionality of the transcript and protein measurements with a consistent pattern of biological changes across both molecular layers for HIF-1 targets.

## Discussion
Severe life-threatening malaria is represented by distinct and overlapping disease features (one or more) of the following: impaired consciousness, prostration, multiple convulsions, acidosis, hypoglycemia, SMA, renal impairment, jaundice, pulmonary edema, significant bleeding, shock, and hyperparasitemia[33]. The clinical manifestations of severe malaria and the age at which they present are largely driven by *P. falciparum* endemicity[8,34]. The overwhelming majority of life-threatening severe malaria occurs in holoendemic *P. falciparum* transmission areas of sub-Saharan Africa in children under five years who develop SMA, making this severe manifestation a leading cause of childhood deaths in such regions[7,8].

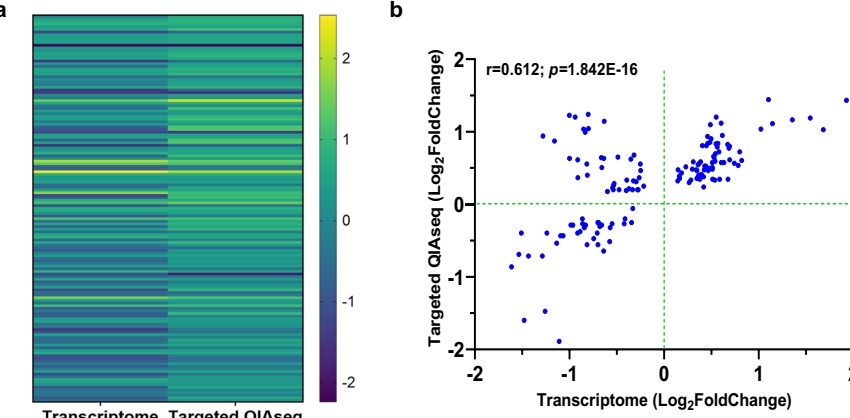

**Fig. 5 | Validation of whole blood transcriptome data using targeted-RNA-Seq panel.** Validation of the RNA-Seq results was performed by comparing the significant ($p < 0.050$) DEGs in the transcriptome analysis with those that were significant in a Qiagen targeted-RNA sequencing panel (491 immune response genes) in a different cohort of Kenyan children [SMA ($n = 21$) and non-SMA ($n = 23$). **a** Heatmap illustrating the comparison of significant ($p < 0.050$) DEGs between the two datasets. The Y-axis depicts the gene pairs, and the X-axis represents the assay type. The color scale depicts fold regulation ($Log_2$). Statistical significance determined using a generalized linear model with a negative binomial distribution, $p < 0.050$. **b** Correlation scatter plot demonstrating the relationship between significantly expressed genes in targeted QIAseq analysis ($Log_2FoldChange$, Y-axis) versus transcriptome data ($Log_2FoldChange$, X-axis). A strong positive correlation of DEGs using two-tailed Spearman's test ($r = 0.612$; 95% confidence interval, 0.496-0.706; $p = 1.842E−16$) validates concordance between the two sequencing methods.

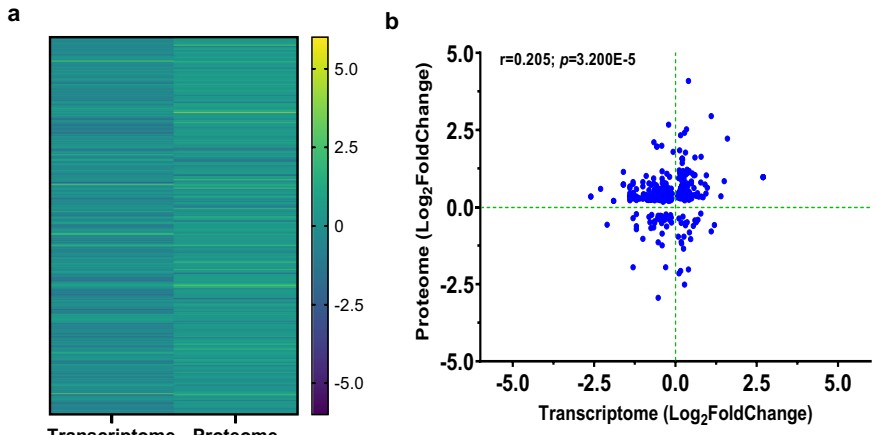

**Fig. 6 | Comparative analysis of transcriptomics and proteome abundance.** To assess the alignment between DEGs identified from RNA-Seq and changes in protein levels, whole blood transcriptome data were compared with protein abundance measured in plasma using a 7k SomaScan platform. MetaCore™ facilitated the mapping of genes to their respective protein products for 35 children ($n = 19$ non-SMA and $n = 16$ SMA) with both available measurements. The analysis identified 405 gene/protein pairs with a significant association ($p < 0.050$). **a** Heatmap showing the comparison of significant ($p < 0.050$) gene/protein pairs between the two datasets. The Y-axis depicts the gene/protein pairs, while the X-axis represents the assay type. The color scale depicts fold regulation ($Log_2$), and $p < 0.050$ calculated using a generalized linear model with a negative binomial distribution. **b** Correlation scatter plot demonstrating the relationship between significantly expressed protein targets ($Log_2FoldChange$; Y-axis) and genes ($Log_2FoldChange$; X-axis). A two-tailed Spearman's test indicated a modest positive concordance between gene expression and protein abundance ($r = 0.205$; 95% confidence interval, 0.107–0.299); $p = 3.200E−5$].

Identifying common gene pathways/networks that encompass the diverse pathophysiological landscape of severe malaria (i.e., mixed phenotype) has presented significant challenges, likely because distinct biological processes may not share common networks. A major advantage of studies in holoendemic malaria regions, such as western Kenya, is that children have a distinct pathophysiological presentation of SMA, making discovery of gene-disease relationships more feasible. Our previous studies have identified innate immune response genes that influence the pathogenesis of SMA, largely through imparting changes in soluble mediators of inflammation[7,8,12,35,36]. However, this is the first investigation to examine the entire expressed peripheral blood transcriptome in children whose primary phenotype of severe disease is SMA. This study identified previously undescribed biological pathways and process networks that converge on perturbations in cellular and

immune stress responses, illustrating that the pathogenesis of SMA is complex and multifaceted.

Differential expression analysis identified both overlapping and unique cellular pathways between non-SMA and SMA groups. Enhanced Th17-type immune responses and activation of signaling cascades that regulate developmental processes and tissue regeneration in the non-SMA group suggest enhanced pathogen defense and an improved capacity for maintaining tissue homeostasis[37–39]. Activation of gene networks involved in cell survival, growth, and differentiation, and responses to external stimuli in children with SMA appear to indicate adaptive responses to stress and damage[40,41].

Identification of molecular patterns by hierarchical clustering of the DEGs revealed unique co-regulated gene clusters that differentiated non-severe from severe disease with distinct WBC and lymphocyte profiles but showed association with neither level of

parasitemia nor sickle cell status. Relationships that emerged from the analysis were then further deciphered by creating canonical process networks. Children with SMA had a set of down-regulated genes (Cluster 1: IRF1↔SUZ12↔IL-1β↔NRF2↔LHX2) that are central to host signaling, cellular differentiation, and defense[42–45]. Decreased expression of the transcription factor, IRF1, and downstream targets (i.e., IL-1β, caspases 1 and 4, and FasR) in the absence of a Nod-like receptor family (i.e., NLRP1, NLRP3, and NLRC4) response suggest an inability to initiate pyroptosis and dysregulation of the inflammasome[46–48]. Up-regulation of the co-regulated sub-network (Cluster 2: TAL1↔LYL1↔BRD4↔FOXO3A↔EKLF) in the SMA group indicates a physiological attempt to overcome hypoxia through enhanced erythropoiesis and metabolic adjusts for restoration of hemostasis and improved oxygen delivery[49–51].

Results from the immune profiling with CIBERSORTx indicate that children with SMA appear to have a diminished capacity for immune memory response, antigen presentation, and immediate pathogen clearance (reduced neutrophils), typifying an immature or improperly regulated immune response[52]. This pattern of expression is consistent with the results obtained from immunological profiles that emerged from the blood transcriptome module repertoire analysis in which SMA was characterized by down-regulation of gene sets for neutrophils, inflammation, interferon, and cytokine/chemokines. Decreased expression of the gene networks in children with SMA identified by the repertoire analysis suggests impairments in innate immune function, stress responses, pyroptosis, autophagy, and antigen processing and presentation, as well as weakened respiratory burst, reduced energy production, and metabolic derangements[53–60]. The pattern of up-regulated gene sets in children with SMA implies an attempt to enhance erythropoiesis and oxygen transport, altered ubiquitination and protein degradation, and enhanced stress response[49,54,61]. The hematological patterns captured by the CBC, although not as specific, parallel results obtained from the immune cell profiling and BloodGen3 analysis. Collectively, children with SMA exhibit a compromised immune system, characterized by diminished memory response, impaired antigen presentation, and reduced pathogen clearance, indicating an overarching theme of an immature or dysregulated immune response alongside metabolic and hematological adaptations.

To gain further insight into the pathogenesis of SMA, we used a combination of functional enrichment analysis platforms (i.e., GO, Reactome, and canonical pathway maps) for the identification of convergent patterns amongst central themes. The central theme in children with SMA with high concordance in the functional enrichment analysis was disruptions in cellular homeostasis and regulatory pathways. This included marked impairments in protein degradation, heme metabolism, cellular clearance mechanisms such as autophagy and efferocytosis, and alterations in metabolic processes. The proper function of protein degradation pathways, including the ubiquitin-proteasome system and autophagy, is essential for cellular health and the prevention of disease[62]. In children with SMA, impaired protein degradation and autophagy could lead to the accumulation of damaged proteins and organelles, contributing to cellular stress and dysfunction, potentially further impairing erythrocyte production or survival, exacerbating anemia[63]. The observed dysregulation in proteasome-mediated activity parallels our earlier studies that were the first to report DEG in ubiquitin-related processes as a feature of SMA[16,64]. Although altered autophagy, to our knowledge, has not been described in human malaria pathogenesis, defects in autophagy are common in an array of other infectious diseases[65]. Disruptions in heme metabolism could also directly exacerbate anemia since impaired heme synthesis could lead to reduced Hb levels, worsening the oxygen-carrying capacity of already low numbers of RBCs in children with SMA[66]. This pathophysiological state in the context of efferocytosis impairments, the process by which dead or dying cells are

cleared by phagocytes, could lead to increased inflammation and tissue damage, potentially impacting the bone marrow's ability to produce new RBCs[67,68]. Our work has consistently demonstrated that Kenyan children with SMA have cytokine/chemokine imbalances that are associated with inefficient erythropoiesis[8,69]. The finding of altered metabolic processes in children with SMA may also explain profoundly low Hb levels since disruptions in metabolic pathways could affect the energy supply and biosynthetic precursors necessary for erythrocyte production, further challenging the compensation for anemic conditions[70].

Biological validation of the whole blood transcriptome data was performed by comparing results in the Kenyan children with those obtained in a cohort of Ugandan children with SMA ($n = 17$, case) and community children ($n = 12$, household)[22]. Functional enrichment analysis of the Ugandan dataset revealed HIF-1 targets as the top-ranked pathway map and the second-ranked pathway map in the Kenyan children. This finding is particularly relevant to SMA since the HIF-1 pathway plays a central role in the response to low oxygen levels[71,72]. For example, activation of the HIF-1 pathway assists in adapting to hypoxic conditions by activating genes involved in erythropoiesis, angiogenesis, and metabolism to enhance oxygen delivery and utilization[71,72]. Enrichment analysis for the two datasets also identified another common pathway map in children with SMA that complements the HIF-1 findings: ROS signaling involved in oxidative stress. Due to the reduced capacity for oxygen transport in SMA, tissues may experience hypoxia, leading to increased production of ROS that can have a dual effect, inducing cellular damage and triggering adaptive responses to improve oxygen delivery and utilization[73]. The protective physiological response involves upregulation of antioxidant defenses to counterbalance the harmful effects of ROS and reduce damage[73]. However, since persistent oxidative stress can exacerbate tissue damage, influencing the progression and severity of anemia by impairing erythrocyte function and lifespan[63], it remains unclear if this central emergent theme in children with SMA is harmful or beneficial. In summary, despite the Ugandan children being older and experiencing a lower level of *P. falciparum* transmission than the children in western Kenya, activation of pathways in response to hypoxic conditions (i.e., HIF-1 and ROS) appears to be a generalizable theme. These results are consistent with upregulated erythroid responses in response to ROS previously reported in the Ugandan children with SMA[22]. However, some patterns in the Kenyan cohort differ such as down- versus up-regulation of activated dendritic cells and down- versus up-regulation of cytokine/chemokine and inflammation responses, perhaps due to lower levels of adaptive immunity in Kenya children who were substantially younger.

Additional validation of the whole blood transcriptome data was performed using a targeted RNA-Seq panel that contained 491 immune response genes. This validation was performed in a separate group of Kenyan children from the same region. For improved biological insight and elimination of extraneous variations, the comparative analysis was conducted for the 147 common significant genes ($p < 0.050$) in both datasets. There was strong concordance between the whole blood transcriptome and targeted RNA-Seq panel, suggesting a high level of reliability and consistency across the different methodological approaches.

Comparing transcriptome data with protein abundance is crucial for understanding how gene expression levels correlate with actual protein production, providing insights into the biological processes and cellular functions in the context of SMA. Moreover, such a comparison accounts for post-transcriptional and post-translational modifications, offering a more comprehensive view of cellular dynamics. The comparative analysis between the transcriptomics data and protein abundance, involving 19 non-SMA and 16 SMA with both measures, uncovered 405 gene/protein pairs with a modest correlation, reinforcing the biological relevance of our findings. The modest

correlation is consistent with other studies showing the weak predictive power of expressed transcripts with their associated proteins, typically $R^2$ values < 10% [74–78]. The top observed alignment between gene expression and protein levels was for HIF-1 targets, reflecting consistency for both the quantitative and directional shifts across these molecular dimensions. Moreover, these results converge with other enrichment platforms utilized in our analyses, highlighting the central theme of responses generated in states of hypoxia.

In conclusion, an unbiased RNA-Seq analysis capturing the entire expressed blood transcriptome identified key molecular aspects that encompass the complex pathogenesis of SMA. These analyses showed that SMA is defined by disruptions in cellular homeostasis and diminished immune responses, with notable impairments in protein degradation and cellular clearance (ubiquitin-related processes and autophagy), and metabolic processes. Strengths of this study include the extensive clinical characterization of the cohort which allowed for the exclusion of co-infections known to influence the immune response [79–81], and a robust sample size. Another strength of the study is the potential generalizability of themes discovered in different cohorts of children with SMA in regions with differing endemicity. Limitations of the study include potential generalizability to other forms of severe malaria, such as cerebral malaria, which is likely a distinct pathogenesis. Additional limitations include the lack of investigation for host-pathogen interactions. We are currently investigating transcriptional changes in *P. falciparum* in the context of the findings presented here to gain a better understanding of how changes in the parasite influence disease severity. Collectively, these findings suggest that SMA was characterized by adaptive yet potentially insufficient responses to stress and damage for the complex interaction between immune and metabolic pathways. Findings from this study underscore the multifaceted nature of SMA and the importance of a holistic understanding for developing targeted therapies to improve clinical outcomes in future work. By uncovering the complex biological underpinnings of SMA, the clinical relevance of this research has the potential to facilitate improved therapeutic strategies, targeted interventions, and ultimately, better health outcomes for affected children.

## Methods
### Study participants
This prospective acute febrile cohort study was conducted at Siaya County Referral Hospital (SCRH), located in a holoendemic *P. falciparum* transmission region in western Kenya where SMA is among the main causes of childhood morbidity and mortality in the community [2,5,24,25,82,83]. Individuals inhabiting the study area are predominantly from the Luo group (> 96%), an ethnically homogeneous population. In the rural region of Siaya County, Kenya, the assignment of sex at birth is influenced by the available medical facilities and the cultural practices prevalent within the community. Local healthcare practitioners often follow traditional methods to assign sex at birth based on physical characteristics. It is important to note that the binary categorization used in this context does not fully capture the spectrum of gender identities recognized by some of the local cultures. This study relies on definitions provided by local health authorities and community leaders, acknowledging that these practices vary across different communities. Female and male children (sex at birth, age 1-59 mos.) presenting at SCRH with symptoms of infectious diseases with the following inclusion criteria were enrolled: temperature ≥37.5 °C (axillary), distance to hospital ≤25 km, parent/guardian willing/able to sign an informed consent and agree to present for day 14 follow-up. Exclusion criteria for the children included: previous hospitalization for any reason, positive malaria RDT test results but negative peripheral parasitemia, an episode of malaria within the past one month, and presentation for non-infectious diseases. Based on these criteria, 577 children were enrolled (03/2017 to 09/2020, Supplementary Fig. S2). At enrollment, demographic and clinical data were collected, and a physical examination was performed. Prior to treatment with antimalarials or other medications, Venipuncture blood samples (3-4 mL) were collected for laboratory measures.

### Ethics statement
The study was approved by the University of New Mexico Institutional Review Board, and the Maseno University Scientific and Ethics Review Committee. Written informed consent was provided by the parents/legal guardians of the study participants.

### Clinical disease definitions
It is important to define anemia based on geographic location because varying environmental (e.g., malaria endemicity, altitude), genetic (e.g., hemoglobinopathies), and dietary factors influence the prevalence and severity of anemia. Regional variations necessitate tailored definitions of anemia to ensure accurate diagnosis and effective treatment. To geographically define anemia categories, we followed a cohort of children ($n = 1654$) over a 36-month period. For the determination of clinical disease definitions based on anemia status, we included 1644 children (3-48 mos.) with robust follow-up data that included over 19,000 Hb measurements for the modeling (Supplementary Fig. S1). Details of the study area have previously been published [25]. A description of the study participants and longitudinal follow-up schedule for the individuals utilized in the analyses were previously described [14]. The analysis presented utilized two cohorts of children recruited and followed with identical parameters across a temporal continuum: cohort 1 (2003–2005; $n = 777$) and cohort 2 (2007–2012; $n = 877$). Briefly, children presenting with suspected malaria infections or reporting for routine vaccinations were recruited at SCRH. Children with varying severities of malarial anemia ($n = 1319$) and aparasitemic controls ($n = 335$) were enrolled following screening for malaria parasites. Exclusion criteria included: children with non-falciparum parasite strains, confirmed cerebral malaria, previously hospitalized for any reason, or had reported use of antimalarial therapy in the two preceding weeks. After enrollment (day 0), children ($n = 1654$) were scheduled for follow-up visits on day 14 (if they were febrile upon enrollment) and quarterly over 36 months. Physical evaluations and laboratory tests required for comprehensive clinical management of the patients were performed at enrollment, day 14, and each acute and quarterly visit [complete blood counts (CBC), malaria parasitemia measures, and evaluation of bacteremia where indicated]. All samples and biological materials were collected before treatment with antimalarials or other medications. Children were treated according to the Ministry of Health-Kenya guidelines that include antimalarials and blood transfusion of children with Hb<5.0 g/dL and/or Hb < 7.0 g/dL in the context of respiratory distress.

Since SMA is among the leading causes of malaria-related mortality in the region, hemoglobin concentrations were averaged for each individual across all visits and correlated with mortality rates. A dynamic programming approach was employed which exhaustively tested all possible splits from 2-way to 10-way [27]. The criterion for the optimal number of splits was the largest κ for which all pairwise chi-square tests between resulting groups were significant at $p < 0.001$. The dynamic programming algorithm exhaustively searched through all possible cut points for hemoglobin to minimize the function $F\kappa$, which is defined as: $F\kappa = -\sum$ (from j = 1 to k) $[-2nj(pj \log(pj) + (1-pj) \log(1-pj))]$, where $pj$ is the proportion of one's (deaths) in segment j, and $nj$ is the number of observations in segment j. This function represents the sum of the negative double products of the number of observations in each segment and the binary log-likelihood for each segment. Data were segmented into k subgroups to minimize the sum. The chi-squared statistic for each potential split was calculated as: $X^2 = F0 - F\kappa$, where $F0$ is the function value for the entire sample. The $p$-value for assessing the significance of each split was obtained from the chi-squared distribution: $p = $ chisqr($X^2$, k − 1). This method

revealed three significantly different ($p < 0.001$) Hb level groups: ≤5.9 g/dL ($n = 62$, mortality fraction=0.53), 5.91-8.09 g/dL ($n = 209$, mortality fraction=0.15), and >8.09 g/dL ($n = 1,373$, mortality fraction=0.03). These independently derived data parallel anemia categories defined from a previous longitudinal birth cohort (0-48 mos.) with 14,317 repeated Hb measurements in the same geographical region: mild anemia (8.0-9.0 g/dL), moderate anemia (6.0-7.9 g/dL), and severe anemia ( < 6.0 g/dL)[28]. As such, children with *P. falciparum* infections (any density parasitemia) were stratified into two groups based on Hb concentrations: Hb≥6.0 g/dL (non-SMA, $n = 39$), and Hb<6.0 g/dL (SMA, $n = 18$). Although the primary outcome of severe malaria in this study was SMA, other clinical complications were also determined. Respiratory distress was defined as: (tachypnea criteria by age); 0-2 mos. (60 breaths per min., bpm), 2-12 mos. (50 bpm), 1-5 yrs. (40 bpm), retractions (in-drawing of the chest wall); grunting; nasal flaring; use of accessory muscles for breathing; and hypoxia ($SpO_2 < 90$[84–86]. Additional definitions included convulsions (tonic-clonic seizures), hypoglycemia (blood glucose levels<2.2 mM), jaundice (yellowing of skin and/or sclera), and thrombocytopenia (platelet count <$150 \times 10^3/mm^3$)[33]. Children with non-SMA were defined as those presenting with a malaria-positive smear from *P. falciparum* parasitemia (of any density) and Hb ≥6.0 g/dL in the absence/presence of other features of severe malaria defined above. Since we have shown that co-infections (i.e., HIV and bacteremia) exacerbate the development of SMA, HIV testing and blood cultures were performed on all children per our published methods[79–81]. Parents/legal guardians of participating children received pre- and post-test HIV&AIDS counseling.

## Laboratory procedures

Upon presentation to hospital, children were screened for presence of malaria parasites using previously published methods[25]. Briefly, heel/finger-prick blood ( < 100 μL) was drawn and used to determine parasitemia and Hb status for initial screening. Giemsa-stained thick and thin blood smears were then prepared and examined for asexual malaria parasites under oil immersion microscopy[25]. The number of *P. falciparum* parasites was determined per 300 leukocytes, and the parasite density was estimated using the total leukocyte count for each patient. Complete blood counts (CBCs) were determined using a DxH 500 hematology analyzer (Beckman-Coulter). HIV-1 status was determined by two rapid serological antibody tests (i.e., Unigold™ and Determine™) and confirmed by HIV-1 proviral DNA PCR tests as previously described[80]. Bacterial cultures were performed on ~1.0 mL of Venipuncture blood collected aseptically, inoculated into pediatric blood culture bottles (Peds Plus, Becton-Dickinson), and incubated in an automated BACTEC 9050 system (Becton-Dickinson) for 5 days. Positive cultures were examined by Gram staining and sub-cultured on blood agar, chocolate agar, or MacConkey agar plates. Bacterial isolates were identified according to standard microbiologic procedures as described previously[81]. To further characterize potential causes of anemia, sickle-cell trait status was determined by alkaline cellulose acetate electrophoresis (Helena BioSciences).

## Study participant selection for RNA-Seq

To select samples for the RNA-Seq from the 577 enrolled study participants, children with malaria were stratified into two groups based on Hb levels (i.e., Hb≥6.0 g/dL and Hb<6.0 g/dL), and then matched according to age and sex. Further selection criteria for the RNA-Seq included omitting children with any detected co-infections, namely HIV-1, and laboratory-confirmed blood-borne bacterial cultures[79–81]. Children with sickle cell disease (HbSS) were also excluded from analysis. All data presented for analysis, and used to generate figures and tables excluded children presenting with HbSS. This selection strategy yielded 39 children with non-SMA (Hb≥6.0 g/dL) and 18 children with

SMA (Hb<6.0 g/dL). Since severe malaria in western Kenya presents as SMA, it was considered the primary outcome variable in RNA-Seq analysis. However, samples of participants with concomitant clinical features of respiratory distress, hypoxia, convulsions, hypoglycemia, jaundice, and thrombocytopenia were included in the analysis.

## RNA isolation, library construction, and sequencing

Approximately 500μL of whole blood collected from venipuncture prior to treatment was stabilized with Trizol® (Thermo Fisher Scientific Inc.), immediately frozen in liquid nitrogen, and then subsequently stored at −80 °C. Total RNA was batch-isolated using E.Z.N.A® Total RNA Kit (Omega Bio-Tek Inc.), treated with RNase-free DNase I (New England Biolabs Inc.), and further processed using RNA Clean & Concentrator (ZYMO Research Corp.). Prior to library preparation and sequencing, RNA degradation and contamination were captured on agarose gels, with purity confirmed using a NanoPhotometer® (IMPLEN). RNA integrity and quantification were measured using the RNA Nano 6000 Assay Kit on a Bioanalyzer 2100 system (Agilent Technologies). RNA quality was assessed using RNA Integrity Number (RIN), and only samples with RIN value of >8 were used for library preparations. For quality control, a predefined exclusion criteria based on RNA quality and read counts was applied, with the error rate set at <1% (GEO accession number GSE255403). To capture the entire expressed transcriptome, an amount of 1 μg RNA/sample was used as input material for sequencing library construction and postglobin mRNA depletion using GLOBINclear™-Human Kit (Thermo Fisher Scientific Inc.). Sequencing libraries were generated using NEBNext® Ultra™ RNA Library Prep Kit for Illumina (New England Biolabs) following the manufacturer's protocol. mRNA was purified from total RNA using poly-T oligo-attached magnetic beads. Fragmentation was performed using divalent cations, with an elevated temperature in NEBNext First Strand Synthesis Reaction Buffer (5X). To synthesize a 1st strand cDNA, random hexamer primer and M-MuLV Reverse Transcriptase (RNase H-) were used. The 2nd strand cDNA was synthesized using DNA Polymerase I and RNase H. Any residual overhangs were transformed into blunt ends using the exonuclease/ polymerase activities. To prepare the DNA fragments for hybridization, NEBNext Adaptor with hairpin loop structure was ligated to the adenylated 3′ ends of the fragments. The resulting cDNA fragments were purified using the AMPure XP system (Beckman Coulter) to select preferential fragments of 150 - 200 bps in length. The USER Enzyme (New England Biolabs) was used with size-selected, adaptor-ligated cDNA at 37 °C for 15 min followed by 5 min at 95 °C before PCR. Amplification was performed with Phusion High-Fidelity DNA polymerase, Universal PCR primers, and Index (X) Primer. The PCR amplicons were purified (AMPure XP system) and library quality was assessed on the Agilent Bioanalyzer 2100 system (Agilent Technologies). Clustering of index-coded samples was performed on a cBot Cluster Generation System using PE Cluster Kit cBot-HS (Illumina Inc.). Triplicate libraries were generated from SMA and non-SMA samples collected at hospital visit (day 0). Sequencing was performed to a depth of >20 million high-quality mappable reads on an Illumina NovaSeq 6000 sequencer (Novogene Corporation Inc.). All the downstream analyses were based on clean data with high quality. None of the samples were excluded from the analysis due to either RNA quality or number of reads obtained.

## Bioinformatics analysis

**Quality control.** Raw data were quality-controlled and filtered using fastp[87]. Clean reads were obtained by removing reads containing adapter, ply-N sequences (N > 10%), and low-quality reads (base quality of >50% bases of the read is ≤5) from raw data. In addition, Q20, Q30, and GC content of the clean data were calculated, and downstream analysis was performed using high-quality clean data (GEO accession number GSE255403).

**Mapping to reference genome.** Reference genome and gene model annotation files were downloaded from a genome website browser (NCBI/UCSC/Ensembl) directly. Sequences were aligned to the human reference genome (GRCh38.p13)[88] using the STAR software version 2.5[89]. With a depth of sequencing covering >20 million mappable reads, the total mapped reads ratio was >70% in the SMA group relative to non-SMA (GEO accession number GSE255403), representing more than half the subjects in the case group.

**Quantification.** HTSeq v0.6.1 was used to generate read counts mapped to each gene[90]. The expected number of Fragments Per Kilobase Million Reads (FPKM) of each gene was calculated according to the gene, gene length, and number of reads mapped to the gene[91]. An FPKM > 1 was set as a threshold for a gene to the considered expressed.

**Differential expression analysis.** Differential expression analysis of the two clinical conditions (non-SMA vs. SMA) was performed using the EdgeR (3.16.5) package, utilizing a negative binomial (NB) based statistical model on RNA count data. The resulting $p$-values were adjusted using the Benjamini-Hochberg procedure to control for the false discovery rate[92]. Genes identified by EdgeR with adjusted $p$-values ($p$adj) of <0.050 were assigned as differentially expressed. To identify the correlation between different genes, samples were clustered using expression level FPKM utilizing the hierarchical clustering distance method.

**Principal component analysis (PCA).** For high-dimensional data reduction, identification of variance, and visualization of groups and trends, PCA was performed on the DEGs in non-SMA and SMA using the factoextra (1.0.7) package in R. The 'prcomp()' function was used to calculate the principal components, with the model data consisting of 57 observations (participants) of 1,682 variables (genes). Demographic (age and sex) and genetic variants (Hb AA and HbAS) data were integrated with the gene annotation file to generate separate 2D plots.

**Leukocytic immune cell profiling**
The relative percentage of different immune cell types/subtypes in whole blood was imputed using CIBERSORTx[29,30]. This analytical tool processes gene expression data from a bulk admixture of different cell types to estimate the abundance of member cell types in a mixed cell population[29]. The curated signature matrix file, LM22, was used as the reference to deconvolute the relative fraction of different cell types in whole blood, resulting in inference of 22 types/subtypes of leukocytes. Imputation of cell-type specific gene expression levels was performed at the sample level with the output presented as the fractional proportion in whole blood for each study participant. The relative proportions of immune cell types were then compared between the non-SMA and SMA groups.

**Module repertoire analysis of DEGs.** To gain insight into coordinated expression of gene groups and their role in biological processes, particularly for immune response genes, modular analysis of blood transcriptome data was analyzed using the BloodGen3Module package in R. BloodGen3 consists of a fixed repertoire of transcriptional modules tailored for the analysis and interpretation of blood transcriptome datasets. The repertoire includes 382 modules with functional annotations, covering >14,000 transcripts, and allows for fingerprint representations. Group comparison analyses were performed using the Welch's-corrected $t$-test ('Groupcomparison') function. For individual-level comparisons, the FDR threshold was 20%.

**Enrichment analysis.** ClusterProfiler[93,94] R package was used to implement the enrichment analysis. GO analysis of DEGs was used to infer functional and biological functions, correcting for gene length bias[93]. Reactome Enrichment Analysis was used to identify pathways

that mapped to biological and cellular networks[95]. Significantly enriched pathways were identified with KEGG using the R package 'ClusterProfiler'[93]. For all analyses, $p$adj < 0.050 were considered significant enrichment. Confirmation and further discovery of the findings were implemented by using MetaCore™ (https://clarivate.com/products/metacore/) to identify DEGs that mapped to GO processes, networks, and pathways.

**Validation of transcriptome profiles**
To perform an independent validation of our findings, we leveraged an independent cohort of Ugandan children (18 mos. to 12 years) in which whole blood transcriptomics was conducted on children with SMA ($n = 17$) and community children (household controls, $n = 12$)[22]. Validation was performed by identifying the top-expressed pathways in MetaCore™ with identical parameters utilized in the analysis of the Kenyan cohort [$\log_2$foldchange = 0.585 (1.5 linear FoldChange) and FDR < 0.050]. Additional validation was performed by utilizing a Qiagen Targeted RNA-Seq panel (491 genes: immune response) conducted on a different set of children with SMA ($n = 20$) and non-SMA ($n = 23$) to corroborate the expression of overlapping genes identified in the transcriptomic analysis.

**Sample selection for QIA-seq analysis.** Study participants for RNA-Seq analysis were selected from a pool of children with malaria ($n = 1,218$, aged 3-36 months) and stratified into two discrete groups of SMA (Hb<6.0 g/dL, Avg, 4.05 g/dL $n = 20$, cases) and non-SMA (Hb≥6.0 g/dL Hb, Avg, 9.96 g/dL, $n = 23$, controls). Exclusion criteria comprised children with co-morbidities such as bacteremia, HIV, and hemoglobinopathies (sickle cell disease), which can affect the anemia outcome. On the first hospital visit before treatment interventions, leukocytes of the children were collected and used for this study.

**RNA isolation, library construction, and sequencing.** To maintain the integrity of RNA, WBC pellets were kept in an equivalent volume of Trizol® LS (Thermo Fisher Scientific Inc.) after collection from the study site at SCRH. Total RNA was extracted using the RNeasy® Plus Micro Kit (Qiagen) according to the manufacturer's protocol (Qiagen) and purified using the RNeasy Micro Kit (Qiagen). The RNA quality was validated using the 260/280 and 260/230 ratios on a NanoPhotometer® (IMPLEN), and the integrity assessed using an automated capillary electrophoresis on an Agilent 2100 Bioanalyzer (Agilent Technologies) according to the manufacturer's protocol. RNA samples with a RIN < 8 were excluded from QIAseq analysis. Total RNA (125 ng) was used to synthesize cDNA using the First Strand Kit reagents (Qiagen) specific for the Targeted Human Inflammation and Immunity Transcriptome Panel (RHS-005Z, Qiagen). Following cDNA synthesis, gene-specific primers with twelve-base random molecular barcodes were added into each unique target strand via single primer extension, allowing quantification of gene expression in each multiplexed sample. A double QIAseq bead clean-up was done for the samples, followed by an eight-cycle limited gene-specific PCR reaction using a gene-specific primer and a universal primer in a total reaction volume of 25 μL. A second QIAseq bead clean-up was done to eliminate excess primer dimers prior to the final universal enrichment and sample-index PCR reaction of 18 cycles. A last universal enrichment index PCR step included the addition of dual unique index primers to allow for the pooling of all samples prior to sequencing. After the final PCR reaction, the QIAseq bead was cleaned to eliminate unincorporated primers. A QIAseq Library Quant Array Kit (Qiagen) was used to quantify the final purified QIAseq libraries. Individual QIAseq targeted RNA libraries were normalized and pooled using equimolar quantities based on the QIAseq Library Quant Array Kit (Qiagen) concentrations. Specifically, the pooled samples were denatured with 0.2 M NaOH and then diluted with hybridization buffer HT1 to a final dilution concentration of 1.2 pM. Each QIAseq Targeted RNA library was normalized to 4 nM with an

average length of about 300 bp. Equal volumes of the equally concentrated libraries were mixed to create the final library pool for sequencing on an Illumina NextSeq500 device. Sequencing run were performed with 1×151 cycles and 8 bp dual index reads. Three independent sequencing reactions were used to complete the Forward Read 1 sequence with QIAseq Read1 Primer I (Qiagen) and then dual multiplexing of indices with Index 1 Read and Index 2 Read Primers (Illumina).

**Data processing and analysis.** The generated FASTQ files were analyzed using QIAseq RNA Quantification pipelines (Qiagen). Trimming of primers prior to read alignment was performed to confirm sequence identity by the internal sequence. Data normalization was performed by obtaining the average unique molecular index (UMI) tag count for the selected reference genes in each sample and in each group to ensure stability across Samples. The $p$-values were calculated using a generalized linear model with a negative binomial distribution of the replicate normalized gene expression data for each gene and expressed as a fold-change (FC > 1.5 and $p < 0.050$) in SMA relative to non-SMA, with $p < 0.05$ indicating significance.

**Differential expression analysis.** Expression analysis using molecular tag counts from RNA-Seq file was done using Targeted RNA Panel Secondary Data Analysis web portal (v1.0, Qiagen). Samples that failed the Data QC step from missing reference genes or suspected DNA contamination quality control were excluded. After normalizing the data with ten reference genes encompassed in the QIAseq Targeted RNA Panels, fold regulation and fold-change were calculated, and $p$-values generated.

### Comparative Analysis of Transcriptomics and Proteomics
For this analysis, the significant ($p < 0.050$) transcripts and proteins were compared.

**Sample selection for proteomic analysis.** To mitigate inter-individual variability in the analysis, we selected plasma samples from the same group of children whose whole blood transcriptome was performed. This strategy resulted in 40 children with plasma samples in sufficient quantities and quality available for proteomic analysis, stratified into SMA (Hb<6.0 g/dL, $n = 18$, cases) and non-SMA (Hb≥6.0 g/dL, $n = 22$, controls).

**Multiplex plasma proteome profiling by microarray detection.** Venipuncture whole blood samples (1.0 to 3.0 mL) were collected and instantly centrifuged (311 ×g) to separate plasma, aliquoted, and stored at −80 °C until use. The samples (no previous freeze-thaw cycles) were analyzed on a 7k SomaScan Assay v4.1 platform (SomaLogic), following the manufacturer's protocol. Briefly, plasma samples were diluted, and SOMAmers synthesized with a fluorophore, photocleavable linker, and biotin. Diluted samples were incubated with dilution-specific SOMAmers attached to streptavidin beads. The bound proteins were tagged with biotin, while the unbound proteins were washed away. To dissociate the photocleavable linker, the mixture was exposed to ultraviolet (UV) light, releasing complexes into the solution. Specific complexes remained bound while non-specific ones dissociated. A polyanionic competitor was introduced to prevent the reformation of non-specific complexes. Subsequently, new streptavidin beads captured the biotinylated proteins and bound SOMAmers. The SOMAmers were released by denaturing the proteins, and fluorophores measured on a microarray chip. The fluorescence intensity, measured in relative fluorescence units (RFU), inferred the quantity of epitope in the original plasma sample[96]. The aptamer-based scan had median limit of detection (LOD) of 125 fM or 5.3 pg/mL[97].

**Data processing and analysis.** Data standardization steps that comprised normalization and calibration were applied to mitigate systematic biases in raw assay following microarray feature aggregation (Supplementary Table S4). The normalization step involved a sample-by-sample adjustment to overall signals within the plasma dilutions, while calibration constituted an overall plate, and SOMAmer-by-SOMAmer adjustments, aimed at decreasing between-plate variability. The final analysis incorporated 35 samples that passed the quality control check, while five samples were excluded due to elevated normalization scale across dilution factors ($n = 3$), high hybridization scale factor (indicating a leak, $n = 1$), or clogged/low volume ($n = 1$). The protein measurements (RFU) were compared between non-SMA and SMA groups using a generalized linear model with a negative binomial distribution. Proteins were matched to their respective transcripts using network algorithms in MetaCore™, and correlation analyses determined using Spearman's test.

### Reporting summary
Further information on research design is available in the Nature Portfolio Reporting Summary linked to this article.

## Data availability
Gene expression data is available in the Gene Expression Omnibus (GEO; https://www.ncbi.nlm.nih.gov/geo/) under the accession number GSE255403. Source data are provided with this paper.

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

## Acknowledgements

The authors gratefully acknowledge the assistance of the University of New Mexico-Kenya team: Nicholas O. Ondiek, Anne A. Ong'ondo, Chrispine W. Ochieng, Everlyne A. Modi, Joan L. A. Ochieng, Duncan O. Njega, Joseph Oduor, Moses Ebungure, Moses Lokorkeju, Rodney B. Mongare, and Vincent Omanje. We are also indebted to all the parents, guardians, and children who participated in the study. The work was supported by National Institutes of Health (NIH) Research Grants R01AI130473 and R01AI51305 (PI: Dr. Douglas J Perkins), NIH Fogarty International Center Grants K43TW011581 (PI: Dr. Samuel B. Anyona), D43TW05884 (DJP, SBA) and D43TW010543 (SBA, DJP), and LANL-LDRD 20150090DR (BHM, DJP).

## Author contributions

SBA: Conducted experiments, data analyses, and co-wrote the manuscript. QC: Project supervision, technical support, data analyses, and manuscript review and editing. SAW: Data analyses and manuscript review and editing. SWO: Data analyses and manuscript review and editing. YG: Data analyses and manuscript review and editing. ER: Technical support and manuscript review and editing. IH: Project supervision, technical support, and manuscript review and editing. COO: Technical support and manuscript review and editing. CO: Project supervision, technical support, and manuscript review and editing. PDS: Project supervision and manuscript review and editing. BHM: Project supervision and manuscript review and editing. CGL: Project supervision and manuscript review and editing. KAS: Data analyses and manuscript review and editing. DJP: Designed study, project supervision, data analyses, and co-wrote manuscript.

## Competing interests

The authors declare no competing interests.

## Additional information

[1]Department of Medical Biochemistry, School of Medicine, Maseno University, Maseno 40105, Kenya. [2]University of New Mexico-Kenya Global Health Programs, Kisumu and Siaya 40100, Kenya. [3]Department of Internal Medicine, Center for Global Health, University of New Mexico, Albuquerque, NM 87131-0001, USA. [4]Department of Biomedical Sciences and Technology, School of Public Health and Community Development, Maseno University, Maseno 40105, Kenya. [5]Department of Public Health Sciences, University of Miami, Miami 33136, USA. [6]Department of Medical Laboratory Sciences, School of Public Health, Biomedical Sciences and Technology, Masinde Muliro University of Science and Technology, Kakamega 50100, Kenya. [7]Department of Emergency Medicine, School of Medicine, University of New Mexico, Albuquerque, NM 87131-0001, USA. [8]Theoretical Biology and Biophysics Group, Theoretical Division, Los Alamos National Laboratory, Los Alamos, NM 87545, USA. [9]Department of Internal Medicine, Division of Translational Informatics, University of New Mexico, Albuquerque, NM 87131-0001, USA. [10]Department Applied Computer and Bio-Sciences, University of Applied Sciences Mittweida, Mittweida 09648, Germany. ✉e-mail: sanyona@maseno.ac.ke; dperkins@salud.unm.edu

