## [Peer Review File · Nature Communications]

Entire Expressed Peripheral Blood Transcriptome in Pediatric Severe Malarial AnemiaReviewers' Comments:

Reviewer #1:

Remarks to the Author:

General:

This manuscript, described by Anyona SB et al, conducted the transcriptome analysis between children with severe malaria anemia (SMA) and without anemia. The authors collected the peripheral blood samples from 25 SMA and 41 non-SMA patients. The collected samples were subjected to the total RNA sequencing analysis in bulk. A total of 3,420 up-regulated and 3,442 down-regulated transcripts in SMA, compared to non-SMA, were identified. These changes collectively indicated that general impairments should take place in SMA in the host genes which are involved in the inflammasome activation, cell death, innate immune response, and the cellular stress responses. For the observed expression changes, qRT-PCR was performed to confirm the data at this part. Based on these results, which I consider should not be sufficiently convincing, the authors concluded that those are the "key molecular themes" in SMA pathogenesis and should be useful for finding potential drug targets. To my knowledge, this particular aspect of severe malaria is not very well studied and the efforts done by the researcher are valuable. However, there are substantial concerns I have to point out for this paper as follows:

Major points:

1 The presented analysis does not represent sufficiently novel insights. A major finding in this study is on the expression changes in the genes of neutrophil responses, autophagy, endosomal pathways, and activation of ubiquitin-related processes and cellular stress responses (line 287-289). However, this is essentially as expected from many other papers of malaria research, also from those on other infectious diseases. For example, there was a previous publication (<https://pubmed.ncbi.nlm.nih.gov/30060095/>) which compared the transcriptomes between severe malaria anemia and cerebral malaria. Even if this paper may be the first study, focusing particularly on SMA, its scientific advance should be limited.

2 The presented results are only from a rather superficial transcriptome analysis and no functional analysis is conducted. With the presented data, it is not clear which part of the observed expression changes are the cause or the consequence of SMA.

3 Protein level analysis is needed to have a more comprehensive view. A recent serum proteome analytical platform, such as Olink or Somalogic may be useful. Results of the routine blood cell analysis should be also considered. These analyses are important when the authors attempt to discover biomarkers for various diagnosis purposes in a clinical setting.

4 Generally, clinical relevance based on the obtained results should be proposed and demonstrated as to its relevance in a more explicit manner.

5 I assume that the authors should have observed substantial diversity between different patients. The analysis on the average or merely statistical base should have a limited power.

Minor points:

6 As the research was done in a holoendemic region with almost certain previous exposure to *Plasmodium falciparum*, I think the authors should describe the possibility of the previous exposure may have influenced on the expression pattern of severe malaria anemia. Considering the medical records of the patients should be essential.

7 Do the strains of *P. falciparum* have any role in the observed pathogenesis? How about the expression profiles of the parasites? I believe that the transcripts from the parasites should be also

included in the dataset. The authors can associate the transcript profiles between those of humans and parasites with each other. Such an analysis should be useful to enrich the findings of this study.

8 Deconvolution analysis is important, but indirect. Single cell analysis is needed to directly detect and analyze the diverse cellular responses.

9 For the manner of the presentation of the results, I feel that the analyses are somehow overlapping and not always helpful for advancing the knowledge. For example, the term analyses by GO and KEGG are somehow analogous, thus could be made more concise. The authors should consider that some of the data should be moved to supplemental data rather than being presented in a main figure.

10 Practically, it is impossible to select targets for the drug development solely based on this information. Further strategy as well as its supporting data should be presented to narrow down the targets.

Reviewer #2:

Remarks to the Author:

This work by Samuel Anyona and colleagues presents the first large scale analysis (to my knowledge) of the human transcriptome in severe malaria anaemia. Severe malarial anaemia is the most common manifestation of severe malaria in high transmission settings, and so it is important to understand its pathogenesis. Transcriptomic studies provide valuable insights into pathogenesis, and potential leads for therapeutic interventions. The study appears to be well-executed and the manuscript is well written with the data clearly presented, and thus this study has the potential to be an important contribution to understanding of malaria pathogenesis. However there are aspects of the paper which I feel need modification and clarification in order for this potential to be fully achieved.

I have two substantial concerns about the approach taken to analysis of this data, which I believe currently undermine the interpretation of the results. Unfortunately addressing these concerns will necessitate substantial reanalysis of the data, but I think this is justified:

1. The Severe Malarial Anaemia (SMA) group currently includes subjects with sickle cell anaemia (HbSS). Sickle cell anaemia is itself a cause of severe anaemia. I do not think it is justified to include subjects with sickle cell anaemia in the SMA group - they should be excluded. I realise this will reduce the sample size, but it is impossible to know how much of the anaemia in these subjects is due to malaria and how much is due to HbSS. The authors present a supplementary analysis, which shows that there are important differences in the results when the subjects with HbSS are excluded. I think the main analysis presented throughout the paper should be based on the subjects without HbSS.

2. The current analysis does not include any attempt to account for the variation in blood leukocyte populations between individuals, and most importantly between groups. This is very important, because at present it is not possible to quantify the extent to which the differences in gene expression profile are due to the differences in the leukocyte populations and the extent to which they are due to changes in gene expression within each leukocyte population. Simply characterising the variation in leukocyte populations using Cibersortx does not solve the problem. There are a variety of approaches to tackling this, which include adjustment for differences in leukocyte population at the stage of undertaking differential expression analysis, or identifying changes in gene expression associated with specific leukocyte populations using a tool like BloodGen3Module.

I am also concerned that the authors suggest that the raw data will be made available on request. This is not in line with community standards. It needs to be uploaded to a public repository and available for scrutiny by reviewers prior to publication

Additional Major Comments

1. The methods section needs additional detail to fully characterise the study population, the RNA-Sequencing, and the analysis:

- a) How was malaria diagnosed / defined for this study
- b) Why does the definition of SMA used in this study differ from the current WHO definition: Severe malarial anaemia: Haemoglobin concentration ≤ 5 g/dL or a haematocrit of $\leq 15\%$ in children < 12 years of age (< 7 g/dL and $< 20\%$, respectively, in adults) with a parasite count $> 10\,000/\mu\text{L}$ (<https://app.magicapp.org/#/guideline/7089>)
- c) Were all children subjected to the same investigations to exclude co-infections? Did they all have blood cultures?
- d) Did any of the children in the non-SMA and SMA groups have other features of severe malaria? Were all of the non-SMA subjects uncomplicated malaria? This detail should be added to methods and table 1
- e) What was the read length of the sequencing?
- f) Were any samples excluded based on the quality of RNA or number of reads obtained?
- g) How was an "expressed gene" defined? How many mapped reads were necessary to consider a gene as being expressed? How many subjects needed to have expression above this threshold for it to be considered as expressed in one group?
- h) Ciberstortx LM22 includes cell types which are not present in blood eg Mast Cells. The authors should curate this reference dataset to remove irrelevant cell types before applying this their own dataset

2. Results

- a) It is important to present (can be supplementary) data showing the outcome of the RNA-sequencing and some basic quantification and quality control metrics. I would like to see: number of reads per subject, number of Human genome mapped reads, number of uniquely mapped reads, number (%) of ribosomal RNA mapped reads, number (%) of globin mapped reads. These should be compared between groups
- b) It would be instructive to show PCA plots, with subjects coloured by group, sex, age, and Hb AA vs AS
- c) Figure 1 A should include annotation of the most significant gene names, and a supplementary table of all differentially expressed genes with P-values and log-FC should be provided to maximise the use of data for the community
- d) Since Table 1 shows substantial differences in WBC and lymphocyte count between groups, it would be important to show these variables in Fig 1C addition to parasitaemia, sickle status, and age
- e) Neutrophils are similar between groups in Table 1, but significantly lower in SMA in the Ciberstortx estimation. This discrepancy is not addressed
- f) There are apparent discrepancies between the top canonical pathways predicted by KEGG and those predicted by Metacore. The authors somewhat cover this up by saying that it illustrates SMA is complex and multifaceted, which is undoubtedly true, but confidence in the predictions would be increased if they can better explain the discrepancies and illustrate overlap between the two methods
- g) Overall there is tendency to overstate the inferences which are possible from the transcriptomic data. The gene expression data is useful for generating hypotheses about molecular pathways and mechanisms, but is fundamentally limited by using bulk analysis of blood, so there is never certainty that the differentially expressed genes are in the same cells. Hence all of the pathway maps that are presented are very much speculation, and it is important to be very cautious in the wording that is used to describe these inferences.
- h) Validation using targeted gene array: RNA-Seq data is now considered a fairly reliable, but if validation is to be performed it is important that it is robust. At present the DEGs from the qRT-PCR array are used for validation. I suggest that the validation should be done by taking the genes which are common between the qRT-PCR array and the RNA-Seq data then looking at correlation between those which are significantly DE in either RNA-Seq or array. I am not convinced what the Metacore analysis adds because it seems inevitable that a qRT-PCR array of 84 ubiquitination genes will return a significant enrichment of ubiquitination pathways. Supplemental Figure 1B does not have any

explanation of the colours of the bars, so I am unclear how to interpret this.

i) Table 1 is missing interquartile ranges throughout

j) Since previous studies have shown the importance of parasite load as a determinant of gene expression, it would be interesting to look at this specifically, within the SMA group

I have not made substantial comments on the discussion because I anticipate that this will change based on reanalysis of the data. However one major limitation of the study which the authors do not really address is that they have not performed any functional validation of their inferences from the gene expression profiles. One could envisage some simple experiments to test some of the hypotheses the authors have generated eg. inability to initiate pyroptosis and dysregulation of the inflammasome, and their manuscript would be substantially strengthened by some choice functional data (although I do not consider this "essential")

Minor Comments:

i) Introduction: The Lee et al Paper (ref 20) suggested SM may be associated with inadequate suppression of type-1 interferon signalling, not with suppression of type 1 interferon signalling

ii) Throughout, the term is "axillary temperature" not "auxillary temperature"

iii) All full gene names should be given at first use, and correct use of italicisation

iv) New data should not be introduced in the discussion section

v) The more data that can be provided as supplementary files the better, from the point of view of allowing the data to be re-used. It would be nice to have full lists of DEGs, genes in each cluster, genes in pathways etc wherever possible

Reviewer #3:

Remarks to the Author:

This manuscript presents whole blood transcriptomes in children to compare profiles associated with anemia (hgb <6, 25 subjects) and those without anemia (hgb ≥ 6, 41 subjects)) to inform a severe anemia pathogenesis model.

This study was conducted in Kenya, study subjects were children who presented with malaria and grouped according to hgb levels, matched for age and sex. They report genes and gene pathways that are associated with each clinical phenotype. GO for neutrophil, autophagy, endosomal pathway and ubiquitin related processes were most enriched in SMA. The note that leucocyte profile in SMA had transcriptional profiles suggesting higher numbers of naïve B cells, Cd8 T cells, CD4 resting memory cells, resting NK cells, monocytes and M2 macs, higher activated dendritic cells, mast cells neutrophils.

Major

There is a huge amount of data presented, and in various ways, volcano plots, genes, GO pathways, figures with Canonical pathway analysis, and KEGG pathway for protein processing in endoplasmic reticulum, however it is not clear how all this data informs a disease model for SMA, no major take home message.

If there was a major pathway that was dysregulated and resulted in more severe anemia, a functional assay to confirm that pathways importance would allow more solid conclusions to their importance.

Other

- "The immune cell type patterns observed indicate that children with SMA have a decreased antigenic response, reduced immune priming, and enhanced polarization towards cellular proliferation and tissue repair."

♣ Can they provide the specific data that underly these conclusions? Did they examine cellular response to antigens for example; what is the exact evidence that the SMA cells are polarized toward

cellular proliferation and tissue repair (not sure what polarization means in this context, could be more specific).

♣ "10 immune cell types were differentially expressed at 127 $p < 0.050$ (Figure 2A). Children with SMA had increased expression of naïve B cells ($p = 9.741E-05$), CD8 T cells ($p = 0.009$)" A more accurate way to report this may be that using transcriptional profiling to identify cell types, there were increased numbers of naïve B cells.

♣ "Endoplasmic Reticulum Quality Control Dysfunction in SMA" Might say instead ER dysfunction in SMA

- A very large number of genes and pathways demonstrated differential transcription between the groups, why was ubiquitination chose to confirm using PCR. Is this pathway the primary message of the paper, or the most important result to inform a pathogenesis model, the stated goal of the analysis? How is differences in ubiquitination involved in SMA pathogenesis?

-

- "Since hierarchical clustering analysis indicated more pronounced gene dysregulation in children with HbSS, the analyses were repeated without these children. This resulted in an identical network of down-regulated genes (cluster 1), but a different set of up-regulated genes (TAL1↔LYL1↔EKLF1↔HMBS↔RHD) that are involved in the regulation and maturation of erythrocytes and Hb production (review, see Love 47)." What is the hypothesis regarding HbSS and SMA and how do these differences in gene expression provide insights into HbSS genotype during malaria or SMA for example?

-

- "Results from the immune profiling with CIBERSORTx indicate that children with SMA have a 253 decreased antigenic response, reduced immune priming, and an enhanced polarization towards cellular 254 proliferation and repair. The hematological patterns captured by the CBC, although not as specific, 255 parallel results obtained from the immune cell profiling. Consistency between the two independent 256 methods supports the reliability of the observed immune alterations in SMA".

- To gain further insight into SMA pathogenesis, we used a combination of functional enrichment analysis platforms to identify convergent patterns amongst central themes. One distinct feature of SMA was neutrophil activation and degranulation. Although not specifically in children with, previous studies suggest that neutrophil activation is more pronounced in severe malaria 20,48-53. An additional molecular convergence among children with SMA was perturbations in autophagy. Altered autophagy, to our knowledge, has not been described in human malaria pathogenesis, but autophagy defects appear common in various other infectious diseases (review, see Deretic 54)" How exactly do these genes sets inform a model of SMA pathogenesis, what is the exact model for more severe anemia ?

Responses to Referees Letter

Reviewer #1 (Remarks to the Author):

This manuscript, described by Anyona SB et al, conducted the transcriptome analysis between children with severe malaria anemia (SMA) and without anemia. The authors collected the peripheral blood samples from 25 SMA and 41 non-SMA patients. The collected samples were subjected to the total RNA sequencing analysis in bulk. A total of 3,420 up-regulated and 3,442 down-regulated transcripts in SMA, compared to non-SMA, were identified. These changes collectively indicated that general impairments should take place in SMA in the host genes which are involved in the inflammasome activation, cell death, innate immune response, and the cellular stress responses. For the observed expression changes, qRT-PCR was performed to confirm the data at this part. Based on these results, which I consider should not be sufficiently convincing, the authors concluded that those are the “key molecular themes” in SMA pathogenesis and should be useful for finding potential drug targets. To my knowledge, this particular aspect of severe malaria is not very well studied and the efforts done by the researcher are valuable. However, there are substantial concerns I have to point out for this paper as follows:

Major points:

Comment 1

The presented analysis does not represent sufficiently novel insights. A major finding in this study is on the expression changes in the genes of neutrophil responses, autophagy, endosomal pathways, and activation of ubiquitin-related processes and cellular stress responses (line 287-289). However, this is essentially as expected from many other papers of malaria research, also from those on other infectious diseases. For example, there was a previous publication (<https://pubmed.ncbi.nlm.nih.gov/30060095/>) which compared the transcriptomes between severe malaria anemia and cerebral malaria. Even if this paper may be the first study, focusing particularly on SMA, its scientific advance should be limited.

Response 1

We thank the reviewers for the comments and issues raised. As the reviewer is aware, this is the first study focusing on SMA, a leading cause of global childhood mortality. The extensively revised manuscript contains many novel findings that have not been previously reported. It is important to note that the molecular basis of disease for regions in which children develop cerebral malaria with or without SMA is completely distinct from that which develops in a region (such as the current study area) in which children exclusively develop SMA and do not have cases of cerebral malaria. This difference is defined by the transmission intensity, and thus, exposure rate to the disease which conditions the rate and extent to which children develop naturally acquired immunity to malaria. As such, although previous studies have explored the transcriptome differences in severe malaria (cerebral malaria versus SMA), this manuscript focuses on children with a distinct clinical phenotype of severe malaria; SMA, whose pathophysiology certainly differs from other severe malaria manifestations.

Please note, that we strengthen the analysis of our transcriptome data by examining the variance and heterogeneity of gene expression in our clinical groups by principal component analysis (PCA), with results revealing the 1st PCA accounts for 36.1% of variance that differentiated the non-SMA and SMA groups. Further, functional interaction and pathway network analysis suggested altered immune responses, metabolic processes, and erythrocyte differentiation in children with SMA. These results are now supported by leukocytic immune cell analysis using CIBERSORTx which indicated that children with SMA have suppressed antigenic responses and reduced immune priming – this is a novel finding for SMA. Additional analysis of the blood transcriptome through immune profiling with the BloodGen3Module repertoire revealed that children with SMA presented with up-regulated erythroid cells and neutrophil activation, amidst suppressed inflammatory responses. Functional enrichment analysis using GO, Reactome, and KEGG revealed disturbances in cellular homeostasis and regulatory pathways (protein degradation, heme metabolism, cellular clearance mechanisms, and efferocytosis) in children with SMA. These results concur with those of the canonical pathway analysis using MetaCore™ that converged on alterations in cellular stress responses, immune modulation, and metabolic changes in children with SMA.

In the revised manuscript, we now validate these findings by comparing our transcriptome results with a transcriptome dataset from a study conducted in Ugandan children. Specifically, we made a comparison between the children with SMA (n=17, cases) and community household children (n=12, controls). The results from these comparative analyses did not include children with cerebral malaria so a valid comparison could be achieved. Thus, the analyses we present from a reanalysis of the outstanding dataset that they established are also novel findings. For example, the enrichment analysis performed with MetaCore™ utilized 323 significant genes ($\text{Log}_2\text{foldchange}=0.585$ and $\text{FDR}<0.050$) and identified the hypoxia-inducible factor 1 (HIF-1) targets in signal transduction as the top emerging pathway. These results corroborate our transcriptome enrichment analysis that also identified the HIF-1 targets pathway, as well as additional commonality for the emergence of the Reactive Oxygen Species (ROS) signaling pathway in oxidative stress in both datasets.

Additional changes in the revised manuscript include validation of our transcriptome findings using a Qiagen-targeted RNA sequencing panel, comprised of 491 immune response genes and conducted in a separate cohort of Kenya children (non-SMA=23 and SMA=20). The heatmap cluster analysis showed concordance in fold-change and directionality, and a strong correlation between the two platforms ($r=0.612$, $p=1.842\text{E-}16$).

To further infer if the differential transcriptome expression profiles align with the functional changes in protein abundance in this cohort of children, we also performed protein quantification using a 7k SomaScan platform. There was a modest positive correlation of 405 significant ($p<0.050$) gene/protein pairs ($r=0.205$, $p=3.200\text{E-}5$), consistent with other such studies comparing transcriptome changes with protein abundance (de Sousa *et al.*, Global signatures of protein and mRNA expression levels. *Molecular BioSystems* 5, 1512-1526 (2009); Vogel, C *et al.*, Insights into the regulation of protein abundance from proteomic and transcriptomic analyses. *Nature Reviews Genetics* 13, 227-232 (2012); Koussounadis, A. *et al.*, Relationship between differentially expressed mRNA and mRNA-protein correlations in a xenograft model system. *Scientific Reports* 5, 10775 (2015); Bauernfeind, A *et al.*, The predictive nature of transcript expression levels on protein expression in adult human brain. *BMC Genomics* 18, 1-11 (2017); and Jiang, W. *et al.*, Analytical Considerations of Large-Scale Aptamer-Based Datasets for Translational Applications. *Cancers* 14, 2227 (2022).

In addition, canonical pathway map analysis using MetaCore™ identified the HIF-1 targets as the most enriched for the gene/protein pair dataset. Collectively, the revised manuscript and extensive additional analyses suggested by the helpful team of reviewers now contains a wealth of novel data. The overall central theme of convergence in both our data and that from Ugandan children is that SMA is defined by activation of gene/protein networks due to states of hypoxia (i.e., HIF-1 and ROS pathways). Thus, we have taken the comments of our contribution to the field noted by the reviewer very seriously and have extensively revised the manuscript to address this valid concern. We believe that the specific focus on SMA provides valuable insights into the pathogenesis of this particular phenotype, which is a significant global health concern in malaria-endemic regions, particularly in high-transmission settings such as western Kenya. The new findings presented expand both their Ugandan findings as well as ours in the revised manuscript.

Comment 2

The presented results are only from a rather superficial transcriptome analysis and no functional analysis is conducted. With the presented data, it is not clear which part of the observed expression changes are the cause or the consequence of SMA.

Response 2

We appreciate the insight from the reviewer and have modified the revised manuscript to contain a more in-depth analysis of the data (as described in detail under Response 1). The extensive additional analyses in the revised manuscript based on the team of reviewers' feedback now provide important steps forward for gaining an improved understanding of the molecular underpinnings of SMA, a complex multifaceted disease process.

We also sincerely acknowledge the reviewer's comments about "cause and consequence". While the study identifies expression changes and protein abundance associated with SMA, it does not establish causality. However, the additional analyses that have been performed revealed new findings related to erythroid activation pathways and changes in heme metabolism in children with SMA, processes that would likely be a consequence of low hemoglobin concentrations in the context of severe anemia. Additionally, responses to

hypoxic conditions and the subsequent activation of HIF-1 and ROS signaling in response to oxidative stress would also likely be a consequence of the disease process. The emergence of responses to hypoxia emerged from both our Kenyan transcriptomics data and in our analysis of the Ugandan dataset (**Fig. 4b and Supplementary Fig. S5**). Conversely, decreased responses for cytokines/chemokines in children with SMA in our dataset appear to indicate a pathophysiological process that does not provide appropriate control of the infection. All these new findings are presented and discussed in detail in the revised manuscript. However, we recognize the limitation of transcriptomics studies with a single timepoint and inference about cause and effect. We are currently scaling up to perform several longitudinal measurements of transcriptomic changes to gain a better understanding of gene networks that are a response versus those that may be contributing to pathogenesis. Despite this limitation, the study contains valuable insights into the transcriptomic landscape of SMA, providing important novel information for further investigation. As such, this study can be considered a valuable contribution to the field that lays the groundwork for further research aimed at unraveling the causal relationships between gene expression/protein abundance changes and SMA.

Comment 3

Protein level analysis is needed to have a more comprehensive view. A recent serum proteome analytical platform, such as Olink or Somalogic may be useful. Results of the routine blood cell analysis should be also considered. These analyses are important when the authors attempt to discover biomarkers for various diagnosis purposes in a clinical setting.

Response 3

We value the reviewer's suggestion to complement the transcriptome analysis with protein-level analysis. As discussed in detail in Response 1, we took this helpful suggestion and performed proteomic profiling using a SomaScan Assay v4.1 platform (SomaLogic). Specifically, comparison of the whole blood transcriptome dataset with protein abundance showed a modest correlation of 405 gene/protein pair (**Fig. 6a and b**), with the canonical pathway analysis for both datasets revealing HIF-1 targets in signal transduction as the top corroborated network (**Supplementary Fig. S6**). The reviewer rightly points out that routine blood cell analysis is essential, particularly in a clinical context. Indeed, the manuscript presents data from our routine clinical profiles (complete blood counts, CBC) in more detail. Further, we performed a deconvolution analysis of immune cell profiling using CIBERSORTx, which showed that children with SMA had increased expression of naïve B cells and CD8 T cells (**Fig. 2a**). To advance these findings even further, we conducted blood-specific gene expression profiling using the BloodGen3Module in R. The findings from the BloodGen3Module analyses suggest that SMA is characterized by the upregulation of erythroid genes, enhanced neutrophil activation, and impaired inflammatory response (**Figs. 2c-e, and Supplementary Table S2**).

Comment 4

Generally, clinical relevance based on the obtained results should be proposed and demonstrated as to its relevance in a more explicit manner.

Response 4

We have added language to the revised manuscript to highlight the clinical relevance of the findings.

Please note, that the identified genes and protein expression changes, while not establishing causality, offer valuable insights into the molecular mechanisms involved in SMA. Understanding these mechanisms is critical for the development of targeted therapeutic interventions. Indeed, specific genes and proteins that are dysregulated in SMA could serve as potential drug targets, and our study provides a basis for further research in this direction. For example, although not directly addressed in this manuscript, we provide gene and protein networks that are central to the pathogenesis of SMA identified using MetaCore™ (Clarivate) (**Fig. 4b, Figs. S5 and S6**). MetaCore™ has multiple features that identify gene/protein targets that are incorporated into the analysis pipeline. Since the manuscript has so much information currently, we aim to develop the potential drug discovery efforts with the robustness required to address this important issue appropriately.

In addition, by identifying differentially expressed genes and proteins associated with SMA, we are also laying the groundwork for the development of diagnostic biomarkers that can be used to improve early diagnosis of SMA in clinical settings, therefore allowing timely interventions and (potentially) better patient prognosis. In the extensive reworking of the manuscript and many different analysis platforms now added, we kindly ask that the

reviewer understand that we cannot present everything in one manuscript due to a limit for how much information is allowable. Our future investigations will focus specifically on therapeutic targets and the discovery of biomarkers.

Comment 5

I assume that the authors should have observed substantial diversity between different patients. The analysis on the average or merely statistical base should have a limited power.

Response 5

We appreciate the reviewer's observation of potential diversity among patients and the associated limitations of analyzing data on an average or statistical basis. To this end, we took the helpful advice of the reviewers and conducted a blood transcriptome module repertoire analysis using BloodGen3Module in R, which compared individual samples, providing a granular understanding of the individual transcriptional profiles at the module level. This allowed us to capture the diversity among patients beyond mere averages or statistical summaries (**Fig. 2e**). This approach enabled us to understand and document individual variations that may exist within our patients, thereby enhancing the robustness and depth of our analysis.

Minor points:

Comment 6

As the research was done in a holoendemic region with almost certain previous exposure to *Plasmodium falciparum*, I think the authors should describe the possibility of the previous exposure may have influenced on the expression pattern of severe malaria anemia. Considering the medical records of the patients should be essential.

Response 6

We are thankful for the reviewer's insight into the potential impact of previous exposure to *Plasmodium falciparum* in a holoendemic region on the expression pattern of SMA. We recognize that repeated exposure to *P. falciparum* can influence the immune responses and gene expression patterns in the host. Since this was a febrile study for which we had the patient's historical medical records, children were excluded if they had been previously hospitalized for any reason. Exclusion also included children who had a positive RDT test but negative peripheral parasitemia (this indicates an ~28-day window in this region due to antigen positivity). Additionally, individuals who had malaria within the past one month, according to the medical records) were excluded from the analyses. It is anticipated that gene expression and protein profiles would return to baseline following therapeutic treatment of malaria infections in this population within the one-month timeframe.

To make the important point raised by the reviewer transparent, the revised manuscript now reads:
Exclusion criteria for the children included: previous hospitalization for any reason, positive malaria RDT test results but negative peripheral parasitemia, an episode of malaria within the past one month, and presentation for non-infectious diseases.

With further regards to exposure, importantly, children in the two groups were matched on age as a key variable. Although one cannot be certain, it is expected that the children in the rural community had very similar exposure rates to infected mosquitoes.

Comment 7

Do the strains of *P. falciparum* have any role in the observed pathogenesis? How about the expression profiles of the parasites? I believe that the transcripts from the parasites should be also included in the dataset. The authors can associate the transcript profiles between those of humans and parasites with each other. Such an analysis should be useful to enrich the findings of this study.

Response 7

We appreciate the reviewer's suggestion to explore the potential role of *P. falciparum* strains in the observed pathogenesis. We acknowledge this as a limitation of our study, as we did not investigate the role of *P. falciparum* strains and parasite expression patterns in our datasets. We recognize the value of exploring host-pathogen interactions to provide valuable insights into the observed pathogenesis of SMA. As such, we are

currently investigating the changes in *P. falciparum* transcriptional profiles in the context of findings presented here to gain a better understanding of how changes in the parasite influence disease severity. However, this is not included in the revised manuscript due to the time-consuming nature of this ongoing work in the context of getting this extensively revised manuscript published in a timely manner.

Comment 8

Deconvolution analysis is important, but indirect. Single-cell analysis is needed to directly detect and analyze the diverse cellular responses.

Response 8

We are thankful that the reviewer regards the use of deconvolution analysis as useful. While we agree that integrating single-cell analysis could offer valuable complementary information to our findings, these data and accompanying samples for the children in this study are not available. This would need to be performed in a different sample set and the results would not be directly comparable to our bulk RNA sequencing data unless that was reperformed in a matched cohort. However, the results of bulk RNA sequencing data do identify shifts in the cellular composition that may be associated with SMA. As noted above, we have also complemented these results with a detailed analysis of the blood transcriptome through the modular gene network analysis performed with BloodGen3. We will certainly incorporate single-cell analysis into our future research efforts.

Comment 9

For the manner of the presentation of the results, I feel that the analyses are somehow overlapping and not always helpful for advancing the knowledge. For example, the term analyses by GO and KEGG are somehow analogous, thus could be made more concise. The authors should consider that some of the data should be moved to supplemental data rather than being presented in a main figure.

Response 9

We appreciate the reviewer's input on the presentation of our results, specifically in terms of the perceived overlap of certain analyses and the suggestion to consider moving some data to supplemental materials. We acknowledge that some of our analyses, such as GO and KEGG, may exhibit overlap in the information they provide. To streamline the presentation of our results, we have moved the KEGG data (**Supplementary Fig. S3**) and other analyses (**Supplementary Fig. S4-S6**) into the supplementary material to avoid redundancy and show more of the overall granularity of the extensive analyses.

Comment 10

Practically, it is impossible to select targets for the drug development solely based on this information. Further strategy as well as its supporting data should be presented to narrow down the targets.

Response 10

We value the reviewer's perspective on the need for a more comprehensive strategy and supporting data for selecting drug development targets based on our research findings. We wish to clarify that our study primarily aimed to identify dysregulated genes and pathways associated with SMA. As noted above, we are currently embarking on a robust effort to discover therapeutic targets and biomarkers that are distinct from the current manuscript.

Reviewer #2 (Remarks to the Author):

This work by Samuel Anyona and colleagues presents the first large scale analysis (to my knowledge) of the human transcriptome in severe malaria anaemia. Severe malarial anaemia is the most common manifestation of severe malaria in high transmission settings, and so it is important to understand its pathogenesis. Transcriptomic studies provide valuable insights into pathogenesis, and potential leads for therapeutic interventions. The study appears to be well-executed and the manuscript is well written with the data clearly presented, and thus this study has the potential to be an important contribution to understanding of malaria pathogenesis. However there are aspects of the paper which I feel need modification and clarification in order for this potential to be fully achieved.

I have two substantial concerns about the approach taken to analysis of this data, which I believe currently undermine the interpretation of the results. Unfortunately addressing these concerns will necessitate substantial reanalysis of the data, but I think this is justified:

Major points:

Comment 1

The Severe Malarial Anaemia (SMA) group currently includes subjects with sickle cell anaemia (HbSS). Sickle cell anaemia is itself a cause of severe anaemia. I do not think it is justified to include subjects with sickle cell anaemia in the SMA group - they should be excluded. I realise this will reduce the sample size, but it is impossible to know how much of the anaemia in these subjects is due to malaria and how much is due to HbSS. The authors present a supplementary analysis, which shows that there are important differences in the results when the subjects with HbSS are excluded. I think the main analysis presented throughout the paper should be based on the subjects without HbSS.

Response 1

The reviewer raises an important point. We understand that the presence of sickle cell anemia could confound the interpretation of the results, and we agree that addressing this concern is vital for the accuracy of our findings. In the initial analysis, we presented a total sample size of 66 study participants, which included 9 children with HbSS. In line with the reviewer's suggestion, we have excluded the 9 children, and present results for 57 study participants (non-SMA, n=39; and SMA, n=18). As the reviewer notes, this meant an entire reanalysis of all data presented. This has now been achieved with substantial effort by our team and considerable delays in publishing this important work. However, the complete reanalysis and additional computational approaches employed substantially improved the findings of this novel work. As such, we thank the reviewer for this suggestion.

Comment 2

The current analysis does not include any attempt to account for the variation in blood leukocyte populations between individuals, and most importantly between groups. This is very important, because at present it is not possible to quantify the extent to which the differences in gene expression profile are due to the differences in the leukocyte populations and the extent to which they are due to changes in gene expression within each leukocyte population. Simply characterizing the variation in leukocyte populations using Cibersortx does not solve the problem. There are a variety of approaches to tackling this, which include adjustment for differences in leukocyte population at the stage of undertaking differential expression analysis, or identifying changes in gene expression associated with specific leukocyte populations using a tool like BloodGen3Module.

Response 2

We acknowledge the significance of accounting for the variation in blood leukocyte populations, as these differences can indeed impact gene expression profiles. To address this concern, we incorporated the reviewer's advice and performed module repertoire analyses using an R package, BloodGen3Module. The fingerprint results of the module repertoire analysis are presented in **Fig. 2C-E** and Supplementary **Table S1**. These data complement the leukocyte population analysis presented using CIBERSORTx (**Figs. 2A and B**) and indicate leukocyte population variation subsets in the observed gene expression differences in SMA. This analysis proved to be very informative – thank you.

Comment 3

I am also concerned that the authors suggest that the raw data will be made available on request. This is not in line with community standards. It needs to be uploaded to a public repository and available for scrutiny by reviewers prior to publication.

Response 3

We apologize for our oversight and the data used in the analysis are now deposited in the Gene Expression Omnibus (<https://www.ncbi.nlm.nih.gov/geo/>) under the GEO accession number **GSE255403**.

Go to

https://urldefense.com/v3/__https://www.ncbi.nlm.nih.gov/geo/query/acc.cgi?acc=GSE255403__;!!KXH1hvEXy

w!bqe9B5rZqWdWPX5RGIsJQ8NMXC0Yggim-bl80wysi2dAwGiJpgv7DPME7kXgdHdYPPpSnyL-8SWUDkQVtCvY8C4w\$

Enter token gxuviaqcbhgxdyj into the box

Additional Major Comments

Comment 4

The methods section needs additional detail to fully characterize the study population, the RNA-Sequencing, and the analysis:

Response 4

We appreciate this valuable feedback from the reviewer. We understand the importance of providing a comprehensive characterization of the study population. In the revised manuscript the Methods Section now contains substantially more detailed information about the inclusion and exclusion criteria for subjects, demographics, clinical parameters, and any relevant medical history, including the presence of conditions such as sickle cell anemia. To provide a more detailed account of our RNA-Seq methodology, we have also included information about sample collection, preparation, and sequencing platforms used. This encompasses RNA extraction techniques, library preparation methods, sequencing instruments, and data quality control steps. Specific sequencing parameters, and statistical/normalization methods employed in data analysis are also explained in more detail. We also recognize the need to provide additional information about the data analysis processes. As such, we now include a detailed and clear description of the bioinformatics pipelines used, software tools, and any specific parameters and thresholds applied in data processing, normalization, and differential gene expression analysis. Additionally, we have described the statistical tests and corrections for multiple testing, and the approach for data visualization.

Comment 5

How was malaria diagnosed / defined for this study?

Response 5

We appreciate the question by the reviewer. This is detailed in the Methods Section of the revised manuscript which now reads:

*Upon presentation to hospital, children were screened for presence of malaria parasites using previously published methods. Briefly, heel/finger-prick blood (<100 μ L) was drawn and used to determine parasitemia and hemoglobin (Hb) status for initial screening. Giemsa-stained thick and thin blood smears were then prepared and examined for asexual malaria parasites under oil immersion microscope. The number of *P. falciparum* parasites was determined per 300 leukocytes, and the parasite density was estimated using the total leukocyte count for each patient. Children with *P. falciparum* infections (any density parasitemia) were stratified into two groups based on Hb concentrations: $Hb \geq 6.0$ g/dL (non-SMA, $n=39$), and $Hb < 6.0$ g/dL (SMA, $n=18$).*

Comment 6

Why does the definition of SMA used in this study differ from the current WHO definition: Severe malarial anaemia: Haemoglobin concentration ≤ 5 g/dL or a haematocrit of $\leq 15\%$ in children < 12 years of age (< 7 g/dL and $< 20\%$, respectively, in adults) with a parasite count $> 10\,000/\mu$ L (<https://app.magicapp.org/#/guideline/7089>).

Response 6

The reviewer raises an important question. Indeed, the definition of SMA by the WHO is $Hb < 5.0$ g/dL. In the data presented in this manuscript, children with *P. falciparum* infections (any density parasitemia) were stratified into two groups based on Hb concentrations: $Hb \geq 6.0$ g/dL (non-SMA, $n=39$), and $Hb < 6.0$ g/dL (SMA, $n=18$). In our definition, we recognize the importance of defining anemia based on geographic location because varying environmental (e.g., malaria endemicity, altitude), genetic (e.g., hemoglobinopathies), and dietary factors influence the prevalence and severity of anemia. To arrive at our definition, we followed a cohort of 1,644 children (3-48 mos.) over a 36-month period and utilized over 19,000 Hb measurements for disease modeling. This work had not previously been presented and is now included in the revised manuscript

(Supplementary Fig. S1). The rationale presented in the revised manuscript and methods for such are as follows:

Since SMA is a leading cause of malaria-related mortality in the region, hemoglobin concentrations were averaged for each individual across all visits and correlated with mortality rates. A dynamic programming approach was employed which exhaustively tested all possible splits from 2-way to 10-way. The criterion for the optimal number of splits was the largest k for which all pairwise chi-square tests between resulting groups were significant at $p < 0.001$. The dynamic programming algorithm exhaustively searched through all possible cutpoints for hemoglobin to minimize the function F_k , which is defined as: $F_k = -\sum_{j=1}^k [-2n_j(p_j \log(p_j) + (1 - p_j) \log(1 - p_j))]$, where p_j is the proportion of ones (deaths) in segment j , and n_j is the number of observations in segment j . This function represents the sum of the negative double products of the number of observations in each segment and the binary log-likelihood for each segment. Data were segmented into k subgroups to minimize the sum. The chi-squared statistic for each potential split was calculated as: $X^2 = F_0 - F_k$, where F_0 is the function value for the entire sample. The p -value for assessing the significance of each split was obtained from the chi-squared distribution: $p = \text{chisqr}(X^2, k - 1)$. This method revealed three significantly different ($p < 0.001$) hemoglobin level groups: ≤ 5.9 g/dL ($n=62$, mortality fraction=0.53), 5.9-8.09 g/dL ($n=209$, mortality fraction=0.15), and > 8.09 g/dL ($n=1,373$, mortality fraction=0.03). These independently derived data parallel anemia categories defined from a previous longitudinal birth cohort (0-48 mos.) with 14,317 repeated Hb measurements in the same geographical region: mild anemia (8.0-9.0 g/dL), moderate anemia (6.0-7.9 g/dL), and severe anemia (< 6.0 g/dL).

Comment 7

Were all children subjected to the same investigations to exclude co-infections? Did they all have blood cultures?

Response 7

We thank the reviewer for the question. To provide enhanced clarity, the revised manuscript states:

HIV-1 status was determined by two rapid serological antibody tests (i.e., Unigold™ and Determine™) and HIV-1 proviral DNA PCR tests as previously described. Bacterial cultures were performed on ~1.0 mL of venipuncture blood collected aseptically, inoculated into pediatric blood culture bottles (Peds Plus, Becton-Dickinson), and incubated in an automated BACTEC 9050 system (Becton-Dickinson) for 5 days. Positive cultures were examined by Gram staining and sub-cultured on blood agar, chocolate agar or MacConkey agar plates. Bacterial isolates were identified according to standard microbiologic procedures as described previously.

Comment 8

Did any of the children in the non-SMA and SMA groups have other features of severe malaria? Were all of the non-SMA subjects uncomplicated malaria? This detail should be added to methods and table 1.

Response 8

The reviewer raises important questions. The primary clinical outcome of severe malaria in the present study was SMA. To provide better clarity for such, we have included the following statement in the revised manuscript:

Although the primary outcome of severe malaria in this study was SMA, other clinical complications were also determined. Respiratory distress was defined as: (tachypnea criteria by age); 0-2 mos. (60 breaths per min., bpm), 2-12 mos. (50 bpm), 1-5 yrs. (40 bpm), retractions (in-drawing of the chest wall); grunting; nasal flaring; use of accessory muscles for breathing; and hypoxia ($SpO_2 < 90\%$). Additional definitions included convulsions (tonic-clonic seizures), hypoglycemia (blood glucose levels < 2.2 mM), jaundice (yellowing of skin and/or sclera), and thrombocytopenia (platelet count $< 150 \times 10^3/mm^3$).

We have also included additional clinical comparisons in **Table 1**. A comparison between the non-SMA and SMA groups showed comparable results for the clinical features, except for respiratory distress, which was significant with a Fisher's exact test, but not significant after multiple test correction using the Bonferroni-Holm

method (familywise error rate, significance level 0.050). This addition suggested by reviewers provides improved clarity of the clinical features of disease in the two groups.

Because the use of a term such as uncomplicated malaria is confusing, all mention of such an opaque term was removed from the revised manuscript.

Comment 9

What was the read length of the sequencing?

Response 9

We thank the reviewer for the question. We have added the following statement in the revised manuscript:

The resulting cDNA fragments were purified using AMPure XP system (Beckman Coulter) to select preferential fragments of 150~200 bps in length.

Comment 10

Were any samples excluded based on the quality of RNA or number of reads obtained?

Response 10

We appreciate the reviewer's inquiry regarding the quality control process for our RNA sequencing data. We have included the following statement in the revised manuscript:

None of the samples were excluded from the analysis due to either RNA quality or number of reads obtained.

Comment 11

How was an "expressed gene" defined? How many mapped reads were necessary to consider a gene as being expressed? How many subjects needed to have expression above this threshold for it to be considered as expressed in one group?

Response 11

We appreciate the reviewer's questions regarding the definition of an 'expressed gene' and the criteria for considering a gene as such in our study. In our analysis, the determination of 'expressed' genes was based on a commonly employed criterion within RNA-Seq studies, where genes were considered expressed if they met a minimum threshold of mapped reads or read counts across the samples. We have inserted the following statements in the revised manuscript to address the reviewer's inquiry:

The expected number of Fragments Per Kilobase Million Reads (FPKM) of each gene was calculated according to gene, the gene length, and number of reads mapped to the gene (Trapnell, C. et al., Transcript assembly and quantification by RNA-Seq reveals unannotated transcripts and isoform switching during cell differentiation. Nature Biotechnology 28, 511-515 (2010). An FPKM>1 was set as a threshold for a gene to be considered expressed.

Comment 12

Ciberstortx LM22 includes cell types which are not present in blood eg Mast Cells. The authors should curate this reference dataset to remove irrelevant cell types before applying their own dataset

Response 12

We appreciate the reviewer's suggestion and did exactly as suggested. The revised manuscript now has the correct results presented (**Fig. 2a and b**).

2. Results

Comment 13

It is important to present (can be supplementary) data showing the outcome of the RNA-sequencing and some basic quantification and quality control metrics. I would like to see: number of reads per subject, number of Human genome mapped reads, number of uniquely mapped reads; these should be compared between groups.

Response 13

We appreciate the reviewer's interest in obtaining additional quantification and quality control metrics from our RNA-sequencing data. To address this request, we have included the mapped reads statistics summary from the sequencing data, all contained within the Gene Expression Omnibus (<https://www.ncbi.nlm.nih.gov/geo/>) under the GEO accession number GSE255403. To further clarify the RNA preparation, we have included the following statement in the Methods Section:

To capture the entire expressed transcriptome, an amount of 1 µg RNA/sample was used as input material for RNA preparation. Sequencing libraries were generated using NEBNext® Ultra™ RNA Library Prep Kit for Illumina (New England Biolabs) following the manufacturer's protocol. mRNA was purified from total RNA using poly-T oligo-attached magnetic beads. Fragmentation was performed using divalent cations, with an elevated temperature in NEBNext First Strand Synthesis Reaction Buffer (5X). To synthesize a 1st strand cDNA, random hexamer primer and M-MuLV Reverse Transcriptase (RNase H-) were used. The 2nd strand cDNA was synthesized using DNA Polymerase I and RNase H. Any residual overhangs were transformed into blunt ends using the exonuclease/ polymerase activities. To prepare the DNA fragments for hybridization, NEBNext Adaptor with hairpin loop structure was ligated to the adenylated 3' ends of the fragments. The resulting cDNA fragments were purified using AMPure XP system (Beckman Coulter) to select preferential fragments of 150~200 bps in length. The USER Enzyme (New England Biolabs) was used with size-selected, adaptor-ligated cDNA at 37 °C for 15 min followed by 5 min at 95 °C before PCR. Amplification was performed with Phusion High-Fidelity DNA polymerase, Universal PCR primers and Index (X) Primer. The PCR amplicons were purified (AMPure XP system) and library quality was assessed on the Agilent Bioanalyzer 2100 system (Agilent Technologies).

Comment 14

It would be instructive to show PCA plots, with subjects coloured by group, sex, age, and Hb AA vs AS.

Response 14

We have incorporated the reviewer's suggestion to enhance the data visualization in our study by including PCA plots with subjects colored by different variables. The PCA plots in which subjects are color-coded by their respective groups (SMA and non-SMA), sex, age and HbAA versus HbAS are now presented in **Supplementary Fig. S2**.

Comment 15

Figure 1 A should include annotation of the most significant gene names, and a supplementary table of all differentially expressed genes with P-values and log-FC should be provided to maximize the use of data for the community.

Response 15

We value the reviewer's suggestion to enhance the interpretability of our results by providing more comprehensive information about the differentially expressed genes. In **Fig. 1a**, we have included annotations of the most significant gene names. In addition, we have provided a list of genes for **Fig. 1** in the **Source data file** that includes the p_{adj} - and \log_2 foldchange values.

Comment 16

Since Table 1 shows substantial differences in WBC and lymphocyte count between groups, it would be important to show these variables in Fig 1C addition to parasitaemia, sickle status, and age

Response 16

We appreciate the reviewer's insightful suggestion to include additional variables, specifically WBC and lymphocyte counts in **Fig. 1c** to provide a more comprehensive representation of the study's clinical and laboratory parameters. We agree that these variables are important for contextualizing our findings and have incorporated the WBC count and lymphocyte count as variables in the revised figure.

Comment 17

Neutrophils are similar between groups in Table 1, but significantly lower in SMA in the Cibersortx estimation. This discrepancy is not addressed.

Response 17

We appreciate the reviewer's observation regarding the discrepancy between neutrophil counts as presented in **Table 1** and the CIBERSORTx estimation (**Fig. 2a and b**). While the Complete Blood Counts (CBC) provides direct and accurate measurement of neutrophil counts in the blood sample at the time of analysis, CIBERSORTx is a computational deconvolution algorithm that indirectly estimates the relative proportions of different cell types, including neutrophils, in a mixed cell population based on gene expression data. As such, CBC yields direct measurements of neutrophil counts, while CIBERSORTx provides estimated proportions of neutrophils based on gene expression data and reference profiles. This would explain the observed variation in our data presented in **Table 1**. More importantly, the helpful inclusion of the module repertoire analyses using the BloodGen3Module, rectified the discrepancy by demonstrating that neutrophil genes were indeed downregulated (such as in CIBERSORTx), but that neutrophil activation was upregulated (**Figs. 2c-e, and Supplementary Table S2**). This is now described in detail in the Discussion Section of the revised manuscript.

Comment 18

There are apparent discrepancies between the top canonical pathways predicted by KEGG and those predicted by Metacore. The authors somewhat cover this up by saying that it illustrates SMA is complex and multifaceted, which is undoubtedly true, but confidence in the predictions would be increased if they can better explain the discrepancies and illustrate overlap between the two methods.

Response 18

Based on reviewers' comments, the KEGG analysis has been moved to the supplementary information and is no longer a primary theme in the manuscript. Please note that the KEGG platform contains many different species, and that the human database is enriched for pathways (approximately 356) in cancer and others, as opposed to those for infectious disease processes. Conversely, MetaCore™ is proprietary and contains specific biological pathways that were established by non-contradictory state-of-the-art knowledge (continuously updated) of the major categories for human metabolism and cell signaling. There are 1000s of such maps in MetaCore™ making it much more robust. However, in the reanalysis presented in the revised manuscript, the results are not discrepant with top KEGG pathways in the supplementary material but rather complementary for mitophagy, autophagy, and ferroptosis.

Comment 19

Overall there is tendency to overstate the inferences which are possible from the transcriptomic data. The gene expression data is useful for generating hypotheses about molecular pathways and mechanisms, but is fundamentally limited by using bulk analysis of blood, so there is never certainty that the differentially expressed genes are in the same cells. Hence all of the pathway maps that are presented are very much speculation, and it is important to be very cautious in the wording that is used to describe these inferences.

Response 19

We appreciate the reviewer's astute observation regarding the limitations with the interpretation of transcriptomic data, especially when derived from bulk analysis of blood samples. We agree that there are inherent challenges in making definitive inferences about molecular pathways and mechanisms from such data. We acknowledge that transcriptomic data, when obtained from bulk analysis of blood samples, provides valuable insights that can generate hypotheses about molecular pathways and potential mechanisms underlying a complex disease such as SMA. We cannot be certain that the differentially expressed genes are expressed within the same cells. This limitation underscores the need for cautious interpretation, and we have reworded the revised manuscript throughout to achieve a balanced and cautious tone.

Comment 20

Validation using targeted gene array: RNA-Seq data is now considered a fairly reliable, but if validation is to be performed it is important that it is robust. At present the DEGs from the qRT-PCR array are used for validation. I suggest that the validation should be done by taking the genes which are common between the qRT-PCR array and the RNA-Seq data then looking at correlation between those which are significantly DE in either

RNA-Seq or array. I am not convinced what the Metacore analysis adds because it seems inevitable that a qRT-PCR array of 84 ubiquitination genes will return a significant enrichment of ubiquitination pathways. Supplemental Figure 1B does not have any explanation of the colours of the bars, so I am unclear how to interpret this.

Response 20

We appreciate the reviewer's feedback regarding the validation of our RNA-Seq data. We have taken a different approach that comprehensively addresses the reviewer team's concerns, while also directly addressing this reviewer's comments. As described above in detail, we performed a reanalysis of the data and presented an independent validation of our findings, leveraging on a cohort of Ugandan children (18 mos. to 12 years) in which whole blood transcriptomic information was available for children with SMA (n=17) and community children (household controls, n=12), omitting children with cerebral malaria presented in their manuscript (Nallandhighal, S. *et al.*, Whole-Blood Transcriptional Signatures Composed of Erythropoietic and NRF2-Regulated Genes Differ Between Cerebral Malaria and Severe Malarial Anemia. *J Infect Dis* 219, 154-164, doi:10.1093/infdis/jiy468 (2019). In this context, MetaCore™ was used more appropriately to find distinct and overlapping pathways between the datasets.

We also performed an additional, more robust, internal validation of the transcriptomics using a Qiagen Targeted RNAseq panel (491 genes: immune response) conducted on a different set of children with SMA (n=20) and non-SMA (n=23). The heatmap cluster analysis showed concordance in fold-change and directionality, and a strong correlation between the two platforms ($r=0.612$, $p=1.842E-16$, **Fig. 5 and b, Supplementary Table S3, Fig. S6**). We have made certain that all the legends and descriptions of such are now clear.

Comment 21

Table 1 is missing interquartile ranges throughout

Response 21

We welcome the reviewer's keen observation. Where indicated, the interquartile (IQR; values shown in bracket) are provided alongside the data points.

Comment 22

Since previous studies have shown the importance of parasite load as a determinant of gene expression, it would be interesting to look at this specifically, within the SMA group.

Response 22

We appreciate the reviewer's suggestion and note that the original manuscript intentionally matched the non-SMA and SMA groups on age, sex, and parasitemia. When suggested to remove the children with HbSS and reperform all the analyses, the parasitemia variable between the non-SMA and SMA groups remained comparable (i.e., not significantly different for both peripheral parasite density and in each of the age-stratified categories, **Table 1**). We also address parasite load in the heatmap presented for the non-SMA and SMA groups (**Fig. 1C**).

Comment 23

I have not made substantial comments on the discussion because I anticipate that this will change based on reanalysis of the data. However one major limitation of the study which the authors do not really address is that they have not performed any functional validation of their inferences from the gene expression profiles. One could envisage some simple experiments to test some of the hypotheses the authors have generated eg. inability to initiate pyroptosis and dysregulation of the inflammasome, and their manuscript would be substantially strengthened by some choice functional data (although I do not consider this "essential").

Response 23

We appreciate the reviewer's recognition of the potential for functional validation to strengthen the inferences. Please note, that we have expanded the analyses substantially based on reviewers' comments and have now included protein abundance measures as an added level of validation for the most relevant gene expression signals detected in children with SMA. We also included internal validation of the transcriptomics data by

performing additional experiments (Targeted RNA-Seq, 491 immune response genes) in a different group of Kenyan children with non-SMA and SMA. We kindly ask that the specific functional pathways for the most relevant findings be included in the follow-up manuscript that will delve into drug targets and biomarkers. Such functional analyses (e.g., pyroptosis and dysregulation of the inflammasome) would fit perfectly into this work.

Minor Comments:

Comment 24

i) Introduction: The Lee et al Paper (ref 20) suggested SM may be associated with inadequate suppression of type-1 interferon signalling, not with suppression of type 1 interferon signalling

Response 24

We appreciate the reviewer's keenness and attention to detail. We have revised the sentence to read:

....*inadequate suppression of type-1 interferon signaling.*

Comment 25

Throughout, the term is "axillary temperature" not "auxillary temperature"

Response 25

We have corrected the grammatical error in the revised manuscript and turned off autocorrect.

Comment 26

All full gene names should be given at first use, and correct use of italicisation

Response 26

We value the reviewer's attention to detail regarding the presentation of gene names and the appropriate use of italicization. While it is not feasible to provide the names of genes in the main manuscript based on word constraints, we have presented a list of all gene names used in the text of the revised manuscript in the **Source data file**.

Comment 27

New data should not be introduced in the discussion section.

Response 27

We welcome the reviewer's guidance regarding the appropriate content for the Discussion Section. We have expunged all new data from the Discussion Section.

Comment 28

The more data that can be provided as supplementary files the better, from the point of view of allowing the data to be re-used. It would be nice to have full lists of DEGs, genes in each cluster, genes in pathways etc wherever possible.

Response 28

We appreciate the reviewer's consideration for data reusability and agree that providing comprehensive supplementary files can enhance the transparency and utility of our research. We are committed to facilitating access to the data. We have included the full lists of DEGs and genes in clusters and pathways in the supplementary files which are now extensive (**Source data file**).

Reviewer #3 (Remarks to the Author):

This manuscript presents whole blood transcriptomes in children to compare profiles associated with anemia

(hgb <6, 25 subjects) and those without anemia (hgb ≥ 6, 41 subjects)) to inform a severe anemia pathogenesis model.

This study was conducted in Kenya, study subjects were children who presented with malaria and grouped according to hgb levels, matched for age and sex. They report genes and gene pathways that are associated with each clinical phenotype. GO for neutrophil, autophagy, endosomal pathway and ubiquitin related processes were most enriched in SMA. The note that leucocyte profile in SMA had transcriptional profiles suggesting higher numbers of naïve B cells, Cd8 T cells, CD4 resting memory cells, resting NK cells, monocytes and M2 macs, higher activated dendritic cells, mast cells neutrophils.

Major Points:

Comment 1

There is a huge amount of data presented, and in various ways, volcano plots, genes, GO pathways, figures with Canonical pathway analysis, and KEGG pathway for protein processing in endoplasmic reticulum, however it is not clear how all this data informs a disease model for SMA, no major take home message.

Response 1

We appreciate the reviewer's honest feedback about her/his position that there is a wealth of data that lacks an important take-home message. To address this issue, ALL the analyses have been redone to address the three reviewers' comments. This includes in-depth analyses (e.g., BloodGen3Module), the addition of protein abundance measures, and two validation analyses, one using an independent cohort with SMA from Ugandan children and another using Targeted RNA-Seq. These additional analyses converge on central themes and the manuscript, although rich with data, now contains a more directed approach for identifying the most relevant gene networks associated with the development of SMA.

Comment 2

If there was a major pathway that was dysregulated and resulted in more severe anemia, a functional assay to confirm that pathways importance would allow more solid conclusions to their importance.

Response 2

While we did not perform a specific functional assay, we did address this important concern by performing all the new analyses to delve into the functionality listed above in Response 1, and with particular relevance, proteomic profiling corroborating the differentially expressed genes (DEGs) using a SomaScan Assay v4.1 platform (SomaLogic).

Minor Points

Comment 3

“The immune cell type patterns observed indicate that children with SMA have a decreased antigenic response, reduced immune priming, and enhanced polarization towards cellular proliferation and tissue repair.”

Can they provide the specific data that underly these conclusions? Did they examine cellular response to antigens for example; what is the exact evidence that the SMA cells are polarized toward cellular proliferation and tissue repair (not sure what polarization means in this context, could be more specific).

Response 3

We appreciate the reviewer's inquiry about the specific data supporting our conclusions related to immune cell type patterns and their implications in SMA. In our manuscript, we did not specifically examine the cellular response to antigens. To address the reviewer's concern, we have revised the terminology to provide more precise descriptions of the transcriptome data presented, particularly for the deconvolution analyses (CIBERSORTx) and its interpretation.

Comment 4

“10 immune cell types were differentially expressed at 127 $p < 0.050$ (Figure 2A). Children with SMA had increased expression of naïve B cells ($p = 9.741E-05$), CD8 T cells ($p = 0.009$)”

A more accurate way to report this may be that using transcriptional profiling to identify cell types, there were increased numbers of naïve B cells.

Response 4

We value the reviewer's suggestion for a more precise reporting of the increased numbers of naïve B cells and CD8 T cells identified through transcriptional profiling (CIBERSORTx). To address the important nomenclature related to the immune cell profiling with CIBERSORTx, we have modified the revised manuscript to now read:

Altered Leukocytic Immune Cell Profiles in SMA: To determine if leukocytic immune profiles differed in children who developed SMA, a bioinformatics approach was implemented using CIBERSORTx. Despite interindividual variability, transcriptional profiling identified five immune cell types were differentially expressed at $p < 0.050$ (Fig. 2a). Children with SMA had increased expression of naïve B cells ($p = 4.524E-04$) and CD8 T cells ($p = 0.026$) (Fig. 2b). In contrast, the SMA group had a lower proportion of expression for memory B cells ($p = 0.035$), activated dendritic cells ($p = 0.007$), and neutrophils ($p = 0.032$) (Fig. 2b). Based on the immune cell type patterns in the expression data, children with SMA appear to have a decreased antigenic response and reduced immune priming.

Comment 5

“Endoplasmic Reticulum Quality Control Dysfunction in SMA” Might say instead ER dysfunction in SMA

Response 5

We appreciate the reviewer's suggestion for a more concise title. Please note that in the reanalysis after removing children with HbSS, different pathways and networks emerged. However, we incorporated the reviewer's suggestion and made sure that our descriptions were more concise and clear.

Comment 6

A very large number of genes and pathways demonstrated differential transcription between the groups, why ubiquitination was chosen to confirm using PCR. Is this pathway the primary message of the paper, or the most important result to inform a pathogenesis model, the stated goal of the analysis? How are differences in ubiquitination involved in SMA pathogenesis?

Response 6

We agree that our choice was not ideal and have extensively modified the revised manuscript and deleted the validation using the qPCR array. As discussed in Response 1 (reviewer 1), we have now employed Targeted RNA-Seq in a separate group of children with non-SMA and SMA. As such, a comparison of our transcriptome results with a transcriptome dataset from a study conducted in Ugandan children with SMA ($n = 17$, cases) and community household children ($n = 12$, controls) identified the hypoxia-inducible factor 1 (HIF-1) targets in signal transduction as the top emerging pathway. These results corroborate our transcriptome enrichment analysis that also identified the HIF-1 targets pathway, as well as additional commonality for the emergence of the Reactive Oxygen Species (ROS) signaling pathway in oxidative stress in both datasets. Additional validation of the whole blood transcriptome dataset with protein abundance using canonical pathway analysis revealed HIF-1 targets in signal transduction as the top corroborated network (**Supplementary Fig. S6**). Therefore, the overall central theme of convergence in both our data and that from Ugandan children is that SMA is defined by activation of gene/protein networks due to states of hypoxia (i.e., HIF-1 and ROS pathways).

Comment 7

“Since hierarchical clustering analysis indicated more pronounced gene dysregulation in children with HbSS, the analyses were repeated without these children. This resulted in an identical network of down-regulated genes (cluster 1), but a different set of up-regulated genes (TAL1↔LYL1↔EKLF1↔HMBS↔RHD) that are involved in the regulation and maturation of erythrocytes and Hb production (review, see Love 47).”

What is the hypothesis regarding HbSS and SMA and how do these differences in gene expression provide insights into HbSS genotype during malaria or SMA for example?

Response 7

Based on reviewers' suggestions, the entire analysis platform was repeated without children with HbSS. As such, this is no longer included in the revised manuscript.

Comment 8

"Results from the immune profiling with CIBERSORTx indicate that children with SMA have a decreased antigenic response, reduced immune priming, and an enhanced polarization towards cellular proliferation and repair. The hematological patterns captured by the CBC, although not as specific, parallel results obtained from the immune cell profiling. Consistency between the two independent methods supports the reliability of the observed immune alterations in SMA".

Response 8

We appreciate the reviewer's recognition of the consistency between the immune profiling with CIBERSORTx and the hematological patterns obtained from the CBC in children with SMA. We have clarified our interpretation and made much more in-depth and robust findings through the use of CIBERSORTx, blood repertoire profiling (Blood Gen3Module), and the relationship with the CBC findings. The convergence and interpretation of the findings based on the three data streams are presented and discussed in detail in the revised manuscript.

Comment 9

"To gain further insight into SMA pathogenesis, we used a combination of functional enrichment analysis platforms to identify convergent patterns amongst central themes. One distinct feature of SMA was neutrophil activation and degranulation. Although not specifically in children with, previous studies suggest that neutrophil activation is more pronounced in severe malaria. An additional molecular convergence among children with SMA was perturbations in autophagy. Altered autophagy, to our knowledge, has not been described in human malaria pathogenesis, but autophagy defects appear common in various other infectious diseases (review, see Deretic 54)"

How exactly do these genes sets inform a model of SMA pathogenesis, what is the exact model for more severe anemia?

Response 9

As the reviewer will see, the revised manuscript incorporated many additional experimental and analytic platforms to come up with convergent themes in SMA. We believe that the improved manner in which these data are presented with address the reviewer's concerns related to our interpretation of the findings.

Reviewers' Comments:

Reviewer #1:

Remarks to the Author:

First of all, I would like to respectfully express my appreciation for the dedicated efforts of the authors, which they have made on the revision of this manuscript. Various aspects of the omics analyses, spanning from in-depth transcriptome features and their population diversity to their protein-expression consequences, have now revealed the comprehensive view of the molecular mechanisms underlying SMA. The related discussion are very much deepened. Collectively, I believe the manuscript has been very much improved. Again, I cordially appreciate the authors' efforts. In fact, this work should have required tremendous amounts of the efforts, especially under the circumstance in a developing country. I sincerely hope this group should lead the cutting-edge genome science among developing countries hereafter. Also, as the authors describe in the manuscript. SMA still imposes substantial burden to a large number of developing countries worldwide. I sincerely hope the continuous efforts of the authors should give a practically useful solution to address this sever problem.

Reviewer #2:

Remarks to the Author:

The authors have done a commendable job addressing the comments of the reviewers and I believe this has substantially improved the manuscript. Overall, this work now presents a well-justified and insightful analysis of host gene expression in severe malarial anemia, which has been validated by comparison with other independent datasets, and as such it is likely to be robust.

I have just a few comments on the revised manuscript:

1. The addition of the information about differing mortality rates across different Hb categories is very helpful to justify a geographically-relevant Hb threshold to define SMA, but this cohort study is not fully described. If there is a separate reference for this study, please cite it. If this is the first time that this study is being described, it needs to be more fully described. How were the children recruited? How often were they sampled? Did they receive any treatments / interventions?

2. The tables of subject characteristics (Table 1 and Table S3) are confusing. The table legends state "Data are presented as median (interquartile range; IQR)" but there are no interquartile ranges presented anywhere in the tables. Please distinguish more clearly between values which are percent and those which are quantities, which values are IQR, and which are standard deviation

3. I do not understand the start of the section "Comparative Cellular Pathways in non-SMA and SMA Reveals Divergent Metabolic and Immune Responses" (lines 124-127). The differentially expressed genes must be from comparison between SMA and non-SMA. There are 1682 DEGs in total. So what what does the next line mean, describing 602 of the DEGs as uniquely expressed in non-SMA, 493 in SMA and 16036 co-expressed? Why do these sum to more than 1682? What is the definition of uniquely expressed in a group? How many subjects have to express the gene in one group but not the other?

4. Line 223. Why is a log2 fold change = 0.585 chosen?

Reviewer #3:

Remarks to the Author:

very responsive, additional cohorts added, message is more clear

Figure 2 may need Y axis label

(not clear if ss patients were completely removed from all the analysis if so then their data should be removed from the figures)

Response to Referees

REVIEWERS' COMMENTS

Reviewer #1 (Remarks to the Author):

Comment 1

First of all, I would like to respectfully express my appreciation for the dedicated efforts of the authors, which they have made on the revision of this manuscript. Various aspects of the omics analyses, spanning from in-depth transcriptome features and their population diversity to their protein-expression consequences, have now revealed the comprehensive view of the molecular mechanisms underlying SMA. The related discussions are very much deepened. Collectively, I believe the manuscript has been very much improved. Again, I cordially appreciate the authors' efforts. In fact, this work should have required tremendous amounts of the efforts, especially under the circumstance in a developing country. I sincerely hope this group should lead the cutting-edge genome science among developing countries hereafter. Also, as the authors describe in the manuscript, SMA still imposes substantial burden to a large number of developing countries worldwide. I sincerely hope the continuous efforts of the authors should give a practically useful solution to address this severe problem.

Response 1

We wish to sincerely thank the reviewer for the thoughtful and appreciative comments regarding our manuscript. We are deeply grateful for the reviewer's recognition of the efforts invested in revising the manuscript. The reviewer's acknowledgment of the depth and breadth of the omics analyses presented in the revised manuscript is encouraging. We are particularly humbled by the reviewer's recognition of the challenges faced in conducting the research in a developing country, and we appreciate the encouragement for our group to continue leading in cutting-edge genome science within this context. Indeed, this work would not have been better without the reviewer's in-depth understanding and valuable comments raised in the initial review. With the reviewer's encouragement, we are motivated to explore how the insights from our research can be translated into practical solutions to alleviate the burden of SMA in developing countries. Once again, we extend our gratitude for the reviewer's constructive feedback and encouragement.

Reviewer #2 (Remarks to the Author):

The authors have done a commendable job addressing the comments of the reviewers and I believe this has substantially improved the manuscript. Overall, this work now presents a well-justified and insightful analysis of host gene expression in severe malarial anemia, which has been validated by comparison with other independent datasets, and as such it is likely to be robust.

I have just a few comments on the revised manuscript:

Comment 1

The addition of the information about differing mortality rates across different Hb categories is very helpful to justify a geographically-relevant Hb threshold to define SMA, but this cohort study is not fully described. If there is a separate reference for this study, please cite it. If this is the first time that this study is being described, it needs to be more fully described. How were the children recruited? How often were they sampled? Did they receive any treatments / interventions?

Response 1

We appreciate the reviewer's comment on the cohort used to justify a geographically relevant Hb threshold to define SMA. Indeed, the cohorts utilized for this analysis have been published previously (Kisia LE, Cheng Q, Raballah E, Munde EO, McMahan BH, Hengartner NW, Ong'echa JM, Chelimo K, Lambert CG, Ouma C, Kempaiah P, Perkins DJ, Schneider KA, Anyona SB. *Genetic variation in CSF2 (5q31.1) is associated with longitudinal susceptibility to pediatric malaria, severe malarial anemia, and all-cause mortality in a high-burden malaria and HIV region of Kenya*. Trop Med Health. 2022 Jun 25;50(1):41. doi: 10.1186/s41182-022-00432-5. PMID: 35752805; PMCID: PMC9233820).

In addition to citing this work in the revised manuscript, we have also added the following:

To geographically define anemia categories, we followed a cohort of children ($n=1,654$) over a 36-month period. For the determination of clinical disease definitions based on anemia status, we included 1,644 children (3-48 mos.) with robust follow-up data that included over 19,000 Hb measurements for the modeling (**Supplementary Fig. S1**). Details of the study area have previously been published (Ong'echa et al., 2006). A description of the study participants and longitudinal follow-up schedule for the individuals utilized in the analyses were previously described (Kisia et al., 2022). The analysis presented utilized two cohorts of children recruited and followed with identical parameters across a temporal continuum: cohort 1 (2003–2005; $n = 777$) and cohort 2 (2007–2012; $n = 877$). Briefly, children presenting with suspected malaria infections or reporting for routine vaccinations were recruited at SCRH. Children with varying severities of malarial anemia ($n = 1,319$) and a parasitemic controls ($n = 335$) were enrolled following screening for malaria parasites. Exclusion criteria included: children with non-falciparum parasite strains, confirmed cerebral malaria, previously hospitalized for any reason, or had reported use of antimalarial therapy in the two preceding weeks. After enrollment (day 0), children ($n = 1654$) were scheduled for follow-up visits on day 14 (if they were febrile upon enrollment) and quarterly over 36 months. Physical evaluations and laboratory tests required for comprehensive clinical management of the patients were performed at enrollment, day 14, and each acute and quarterly visit [complete blood counts (CBC), malaria parasitemia measures, and evaluation of bacteremia where indicated]. All samples and biological materials were collected before treatment with antimalarials or other medications. Children were treated according to the Ministry of Health-Kenya guidelines that include antimalarials and blood transfusion of children with $Hb < 5.0$ g/dL and/or $Hb < 7.0$ g/dL in the context of respiratory distress.

Comment 2

The tables of subject characteristics (Table 1 and Table S3) are confusing. The table legends state "Data are presented as median (interquartile range; IQR)" but there are no interquartile ranges presented anywhere in the tables. Please distinguish more clearly between values which are percent and those which are quantities, which values are IQR, and which are standard deviation

Response 2

We thank the reviewer for raising an important query. We have revised Table 1 and Table S3 legends to indicate clearly what values are presented. For example, the editing includes the relationship between the statistical test and the type of data presented (i.e., ^aFisher's exact test [presented as number (%)], ^bTwo-sided Mann-Whitney-U tests [presented as median (IQR)], and ^cGroup means compared by two-sample t-test [presented as mean (SEM)]). We have revised the legends as follows:

Data are presented as number (percentages; %), median (interquartile range; IQR) or mean (standard error of mean; SEM) unless otherwise noted. Children ($n=57$) presenting with malaria at SCRH were recruited. Based on hemoglobin (Hb) levels, children were categorized into either non-severe malarial anemia (non-SMA; $Hb \geq 6.0$ g/dL, $n=39$) or severe malarial anemia (SMA; $Hb < 6.0$ g/dL, $n=18$). ^aFisher's exact test [presented as number (%)] with exact p -values for homogeneity was performed. ^bTwo-sided Mann-Whitney-U tests [presented as median (IQR)] were used to compare the non-SMA and SMA groups, ^cGroup means were compared by two-sample t -test [presented as mean (SEM)], with equal variance.

Comment 3

I do not understand the start of the section "Comparative Cellular Pathways in non-SMA and SMA Reveals Divergent Metabolic and Immune Responses" (lines 124-127). The differentially expressed genes must be from comparison between SMA and non-SMA. There are 1682 DEGs in total. So what does the next line mean, describing 602 of the DEGs as uniquely expressed in non-SMA, 493 in SMA and 16036 co-expressed? Why do this sum to more than 1682? What is the definition of uniquely expressed in a group? How many subjects have to express the gene in one group but not the other?

Response 3

The reviewer raises an important point. Indeed, the divergence in the number of genes presented in Figures 1a and 1b requires clarification. Fig1a identified significant DEGs ($n=1682$) from our entire set of 53,286 transcripts quantified. Fig1b presents a Venn diagram analysis between non-SMA and SMA derived from all the 53,286 transcripts measured. To provide more clarity, we have revised and reorganized (changed the sequence of the Figures – Fig1b is now Fig1a). The revised manuscript now reads:

Comparative Cellular Pathways in non-SMA and SMA Reveals Divergent Metabolic and Immune Responses: To identify unique and shared genes, a Venn diagram analysis was performed on the 53,286 transcripts, revealing 602 genes that were uniquely expressed in non-SMA, 493 in SMA, and 16,036 co-expressed genes (**Fig. 1a**). Uniquely expressed genes in non-SMA and SMA were then explored by canonical pathway analysis (MetaCore™), revealing four significant sub-networks (**Supplementary Table S1**). The top-ranked sub-network in children with non-SMA was [TFF3↔IL-6↔IL6RA↔ADAM17↔gp130, ($p=3.180E-12$)], highlighting gene ontology (GO) processes for the crucial role of T-helper 17 cell lineage commitment and differentiation, interleukin-6-mediated signaling, and T-helper cell lineage commitment in driving T-helper 17 type immune responses. The second sub-network [SHOX2↔Neuregulin1↔FGFR2↔Endothelin-1↔ECE2, ($p=1.930E-07$)] indicates processes that regulate cellular signaling and proliferation, particularly through receptor tyrosine kinase and enzyme-linked pathways. The top-ranked sub-network in children with SMA [c-Myc↔C/EBPbeta↔STAT1↔STAT5↔ERK1/2, ($p=1.240E-200$)] was associated with signaling pathways linked to growth factors, peptide hormones, stimuli, and transmembrane receptor protein tyrosine kinases. The second sub-network [ACE1↔des-Arg9-bradykinin↔BDKRB1↔des-Arg10-kallidin↔BDKRB2, ($p=5.310E-103$)] underscores positive regulation of cellular responses associated with hormones, chemicals, and metabolic processes. Collectively, these data indicate that non-SMA was characterized by changes in immune response and cell signaling, while SMA involved a broader spectrum of cellular activities, including metabolic regulation and hormone responses. Differential expression analysis of the 53,286 transcripts was then performed, identifying 1,682 DEGs [false discovery rate (FDR)-adjusted, $padj<0.050$]: 1,403 up- and 279 down-regulated genes in children with SMA (**Fig. 1b**).

Comment 4

Line 223. Why is a \log_2 fold change = 0.585 chosen?

Response 4

We thank the reviewer for the question raised. We have clarified the meaning in the revised manuscript. The \log_2 of 1.5 is equal to 0.585 or 2 raised to the power of 0.585 is approximately 1.5. We selected the standard value of 1.5 foldchange ($\log_2=0.585$) to include a threshold that could balance sensitivity and specificity in identifying differentially expressed genes. With a more moderate threshold of 0.585, we aimed to make discoveries in a biological context, yet exclude too many false discoveries.

The manuscript has been revised for clarity and now reads:

*Additional characterization of SMA pathogenesis was carried out by exploring the top 10 significant canonical pathway maps with a \log_2 foldchange=0.585 (1.5 linear foldchange) and false discovery rate (FDR)<0.050 (MetaCore™, **Fig. 4a**).*

Reviewer #3 (Remarks to the Author):

Very responsive, additional cohorts added, message is more clear

Comment 1

Figure 2 may need Y axis label

Response 1

We note the reviewer's keen attention and agree that there needs to be a Y axis label in Figure 2b. We have added the label in the revised Figure.

Comment 2

(Not clear if ss patients were completely removed from all the analysis if so then their data should be removed from the figures).

Response 2

We appreciate the reviewer's observation. We wish to state that all patients with HbSS were excluded from all analyses, and the figures and tables presented in the revised manuscript were recreated afresh with data that excluded HbSS. We have added the following statement in the revised manuscript:

All data presented for analysis and used to generate figures and tables excluded children presenting with HbSS.